# TIME WILL TELL: NEW OUTLOOKS AND A BASELINE FOR TEMPORAL MULTI-VIEW 3D OBJECT DETECTION

**Jinhyung Park**[1*]   **Chenfeng Xu**[2*]   **Shijia Yang**[2]   **Kurt Keutzer**[2]   **Kris Kitani**[1]
**Masayoshi Tomizuka**[2]   **Wei Zhan**[2]
[1]Carnegie Mellon University   [2]University of California, Berkeley
{jinhyun1, kkitani}@cs.cmu.edu
{xuchenfeng, shijiayang, keutzer, tomizuka, wzhan}@berkeley.edu

## ABSTRACT

While recent camera-only 3D detection methods leverage multiple timesteps, the limited history they use significantly hampers the extent to which temporal fusion can improve object perception. Observing that existing works' fusion of multi-frame images are instances of temporal stereo matching, we find that performance is hindered by the interplay between 1) the low granularity of matching resolution and 2) the sub-optimal multi-view setup produced by limited history usage. Our theoretical and empirical analysis demonstrates that the optimal temporal difference between views varies significantly for different pixels and depths, making it necessary to fuse many timesteps over long-term history. Building on our investigation, we propose to generate a cost volume from a long history of image observations, compensating for the coarse but efficient matching resolution with a more optimal multi-view matching setup. Further, we augment the per-frame monocular depth predictions used for long-term, coarse matching with short-term, fine-grained matching and find that long and short term temporal fusion are highly complementary. While maintaining high efficiency, our framework sets new state-of-the-art on nuScenes, achieving first place on the test set and outperforming previous best art by 5.2% mAP and 3.7% NDS on the validation set. Code will be released here: https://github.com/Divadi/SOLOFusion.

## 1 INTRODUCTION

Recent advances in camera-only 3D detection have alleviated monocular ambiguities by leveraging a short history. Despite their improvements, these outdoor works neglect the majority of past observations, limiting their temporal fusion to a few frames in a short 2-3 second window. These long-term past observations are critical for better depth estimation, which has been demonstrated through oracle experiments (Wang et al., 2021b; Jing et al., 2022) as the main bottleneck of camera-only pipelines due to their lack of explicit depth measurements.

Although existing methods aggregate temporal features differently, in essence, these works all consider regions in 3D space and consider image features corresponding to these hypothesis locations from multiple timesteps. Then, they use this temporal information to determine the occupancy of or the existence of an object at those regions. As such, these works are instances of *temporal stereo matching*. To quantify the quality of multi-view (temporal) depth estimation possible in these methods, we define *localization potential* at a 3D location as the magnitude of the change in the source-view projection induced by a change in depth in the reference view. As shown in Figure 1, a larger localization potential causes depth hypotheses (Yao et al., 2018) for a reference view pixel to be projected further apart, giving them more distinct source view features. Then, the correct depth hypothesis with a stronger match with the source view feature can more easily suppress incorrect depth hypotheses with clearly unrelated features, allowing for more accurate depth estimation.

We evaluate the localization potential in driving scenarios and find that only using a few recent frames heavily limits the localization potential, and thus the depth estimation potential, of existing methods. Distinct from the intuition in both indoor works, which select frames with above a minimum translation and rotation (Hou et al., 2019; Sun et al., 2021), and outdoor works, which often empirically select a single historical frame (Huang & Huang, 2022; Wang et al., 2022c; Liu et al., 2022b), we find that *the optimal rotation and temporal difference* between the reference and

---

*Equal contribution.

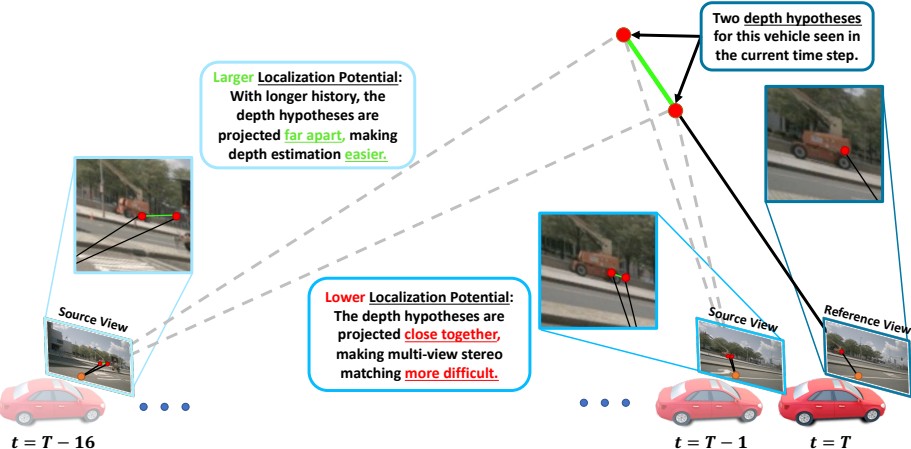

Figure 1: The depth hypothesis projections onto the $t = T - 16$ source view are further apart, making multi-view depth estimation easier when compared to the $t = T - 1$ source view.

source frame *varies significantly over different pixels, depths, cameras, and ego-motion*. Hence, it is necessary to utilize many timesteps over a long history for each pixel and depth to have access to a setup that maximizes its localization potential. Further, we find that localization potential is not only decreased by fewer timesteps but is also hurt by the lower image feature resolution used in existing methods. Both factors significantly hinder the benefits of temporal fusion in prior works.

We verify our theoretical analysis by designing a model that naturally follows from our findings. Although existing methods' usage of low-resolution image feature maps for multi-view stereo limits matching quality, our proposed long-term temporal fusion's dramatic increase in localization potential can offset this limitation. Our model adopts the coarse but efficient low-resolution feature maps and leverages a 16-frame BEV cost volume. We find that such a framework already outperforms prior-arts, highlighting the significant gap in utilizing temporal information in existing literature. We extend our model by further exploiting short-term temporal fusion with an efficient sampling module, replacing monocular depth priors in the 16-frame BEV cost volume with a two-view depth prior. This time offsetting the temporal decrease in localization potential with an increase in feature map resolution, we observe a further boost in performance, demonstrating that **short-term and long-term temporal fusion are highly complementary**. Our main contributions are as follows:

- We define *localization potential* to measure the ease of multi-view depth estimation and use it to theoretically and empirically demonstrate that the optimal rotation and temporal difference between reference and source cameras for multi-view stereo varies significantly over pixels and depths. This runs contrary to intuition in existing works that impose a minimum view change threshold or empirically search for a single past frame to fuse.

- We verify our theoretical analysis by designing a model, **SOLOFusion**, that leverages both **ShO**rt-term, high-resolution and **LO**ng-term, low-resolution temporal stereo for depth estimation. Critically, we are the first, to the best of our knowledge, to balance the impacts of spatial resolution and temporal difference on localization potential and use it to design an efficient but strong temporal multi-view 3D detector in the autonomous driving task.

- Our framework significantly outperforms state-of-the-art methods in utilizing temporal information, demonstrating considerable improvement in mAP and mATE over a strong non-temporal baseline as shown in Figure 2. SOLOFusion achieves first on the nuScenes test set and outperforms previous best art by 5.2% mAP and 3.7% NDS on the validation set.

## 2 RELATED WORK

### 2.1 SINGLE-VIEW CAMERA-ONLY 3D OBJECT DETECTION

Many single-view methods use mature 2D CNNs and predict 3D boxes from the image (Mousavian et al., 2017; Brazil & Liu, 2019; Qin et al., 2019; Xu & Chen, 2018; Zhou et al., 2019). Some works leverage CAD models (Liu et al., 2021; Manhardt et al., 2019; Barabanau et al., 2020) while others set prediction targets as keypoints (Li et al., 2022e; Zhang et al., 2021) or disentangled 3D parameters (Simonelli et al., 2019; Wang et al., 2021a). Another line of work predicts in 3D, using monocular depth prediction networks (Fu et al., 2018; Godard et al., 2017) to generate pseudo-LiDAR (Wang et al., 2019; Weng & Kitani, 2019) and applying LiDAR-based 3D detection frameworks. Our paper addresses monocular 3D ambiguity through temporal fusion and is perpendicular.

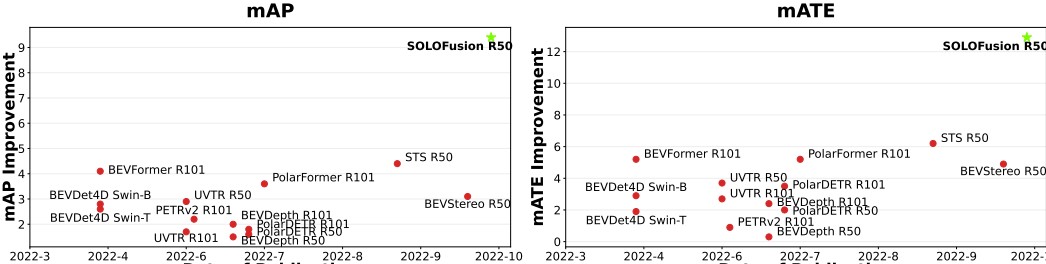

Figure 2: We visualize the improvement of temporal models over their non-temporal baselines, using results from publications and released code. With short-term and long-term temporal fusion, our method best leverages temporal information, demonstrating 9.4% and 12.9% improvements in precision-recall (mAP) and localization quality (mATE), respectively.

## 2.2 MULTI-VIEW CAMERA-ONLY 3D OBJECT DETECTION

Most multi-camera methods operate in 3D space and are divided into grid and query based works. Following LSS (Philion & Fidler, 2020), some grid-based methods predict a distribution over depth bins and generate a point cloud with probability-weighted image features to be used for BEV detection(Reading et al., 2021; Huang et al., 2021; Li et al., 2022a). Followup works (Liu et al., 2022c; Li et al., 2022c) speed up voxelization and introduce depth supervision (Li et al., 2022c). While these methods generate point clouds from images and voxelize, other grid-based works instead project voxels onto images and sample features (Wang et al., 2022b;a; Roddick et al., 2018; Rukhovich et al., 2022). Another branch of works follow DETR3D Wang et al. (2022d) and adopt queries. These works use object queries (Wang et al., 2022d; Liu et al., 2022a; Chen et al., 2022; Jiang et al., 2022) or BEV grid queries (Li et al., 2022d) and project them to get image features.

Recent works extend grid and query frameworks to process several frames. Grid methods align and concatenate volumes from multiple timesteps (Huang & Huang, 2022; Li et al., 2022c;a) or sample voxel features from past images (Wang et al., 2022b;a; Rukhovich et al., 2022). Query-based methods sample from past images through projection (Chen et al., 2022; Jiang et al., 2022; Qin et al., 2022) or attention (Liu et al., 2022b; Li et al., 2022d). However, these methods demonstrate limited improvement from temporal fusion. Most of these works fuse multi-frame features at low resolution and use a limited number of timesteps over a short time window - both factors that we find severely decrease localization potential. In our work, we quantify these limitations and propose a strong framework that explicitly considers the relationship between spatial resolution and temporal history.

## 2.3 MULTI-VIEW STEREO

Multi-view stereo works use depth maps (Kang et al., 2001) and 3D volumes (Kutulakos & Seitz, 2000) to conduct depth, mesh, and point-cloud reconstruction tasks. Several methods (Galliani et al., 2015; Ji et al., 2017) use a depth map fusion algorithm for large-scale scenes. Other works (Kar et al., 2017; Yao et al., 2018; Yang et al., 2020; Zbontar et al., 2016; Yao et al., 2019) generate a volume by scattering pixel features to the 3D grid and estimating occupancy cost for each voxel. A few methods apply multi-view stereo works to 3D detection. DfM (Wang et al., 2022b) generates a plane-sweep volume from consecutive frames. STS (Wang et al., 2022e) uses spatially-increasing discretization (SID) depth bins (Fu et al., 2018) for stereo matching, and BEVStereo (Li et al., 2022b) adapts MaGNet (Bae et al., 2022) and dynamically selects candidates for iterative multi-view matching.

Although these methods demonstrate further improvement, the short history they use for detection limits their gain from temporal fusion as shown in Figure 2. We formulate and analyze the connection between temporal camera-only 3D detection and multi-view stereo and verify our analysis by leveraging the synergy of short-term and long-term fusion. Further, our proposed short-term temporal fusion is more efficient and extensible, and it also demonstrates larger improvement.

## 3 UNIFIED TEMPORAL STEREO FORMULATION FOR CAMERA-ONLY 3D DETECTION

### 3.1 BACKGROUND OF MULTI-VIEW STEREO MATCHING

Multi-view stereo matching estimates the depth map of an image (reference view) by leveraging additional images taken of the same scene (source views). For each pixel in the reference view, multi-view stereo frameworks consider many depth hypotheses (Yao et al., 2018) - locations this pixel could be in the 3D world. Of these hypotheses, the true depth of the pixel appears most similar through the reference and source views. Thus, each depth hypothesis is projected onto the source

Table 1: *Sampled image features are features weighted by probability prediction for that depth. †STS and BEVStereo also perform the BEV temporal stereo done in BEVDepth. ‡BEVFormer aggregates previous timestep features recurrently. They set the previous time window to be 3 timesteps during training and maintain a running BEV feature map during inference.

| Type | Method | Candidate Loc. | Sampling Op. | Sampling Res. | Temporal Agg. | Prev Time | Cand. Loc. Proc. | Task/Supervision |
|---|---|---|---|---|---|---|---|---|
| MVS | MVSNet | plane-sweep volume | projection & bilinear | 1/4 | variance | 2; - | 3D conv | depth estimation/L1 |
| | MaGNet | predicted gaussian confidence interval | projection & bilinear | 1/4 | dot product | 2 or 4; - | 2D conv | depth estimation/L2 |
| Grid-Based | BEVDet4D | BEV grid cells | image feats* BEV pool | 1/16 | align & concat | 1; 2.5s | 2D conv | obj. pred. & localization |
| | BEVDepth | BEV grid cells | image feats* BEV pool | 1/16 | align & concat | 1; 0.5s | 2D conv | obj. pred. & localization |
| | STS† | SID plane-sweep vol. | projection & bilinear | 1/4 | groupwise corr. | 1; 0.5s | MLP | depth estimation/BCE |
| | BEVStereo† | predicted gaussian confidence interval | projection & bilinear | 1/4 | groupwise corr. | 1; 0.5s | MLP | depth estimation/BCE |
| | DfM | plane-sweep vol. | projection* & bilinear | 1/1 | concat | 1; 0.3s | 3D & 2D conv | depth est. & obj. pred. & loc. |
| | MV-FCOS3D++ | 3D grid cells | projection & bilinear | 1/4 | concat | 1; 1s | 3D & 2D conv | obj. pred & localization |
| Query-Based | BEVFormer | BEV query loc. | projection & deform attn | 1/16 - 1/64 | align & deform attn | 3; 0 - 2s‡ | trans. decoder | obj. pred. & localization |
| | UniFormer | BEV query loc. | projection & bilinear | 1/8 - 1/64 | weighted sum | 6; 0 - 3s | trans. decoder | map segmentation |
| | PolarDETR | object query loc. | projection & deform attn | unspecified | concat | 1; 0.5s | trans. decoder | obj. pred. & localization |
| | PolarFormer | object query loc. | deform attn onto image feature polar BEV map | unspecified | align & concat | 1; 2.5s | trans. decoder | obj pred. & localization |
| | UVTR | object query loc. | deform attn onto image feature* polar 3D volume | 1/4 - 1/32 | align & concat | 5; 0 - 1s | obj. trans. decoder | obj. pred. & localization |
| | PETRv2 | object query loc. | cross-attention | 1/16 | align & cross-attn | 1; 2.5s | trans. decoder | obj. pred. & localization |

views, and the depth with the best image feature match between the source view projection and the reference pixel is selected as the depth estimate. In the standard stereo setting, the reference and source views are left and right cameras with synchronized, rectified images. For temporal stereo, the reference and source views are images captured at the current and past timesteps as illustrated in Figure 1. For additional details on multi-view stereo, we refer readers to MVSNet (Yao et al., 2018).

## 3.2 COMPONENTS OF A UNIFIED FORMULATION

Camera-only 3D detection methods using multiple frames each propose their own method for temporal feature aggregation. However, we find that these works are mostly grounded in the same core formulation of multi-view stereo matching. We identify the main components of such a formulation:

- Candidate Locations - the locations in 3D space considered for matching.
- Sampling Operation - the method used to obtain image features for a candidate region.
- Sampling Resolution - the spatial resolution of the image features used for sampling.
- Temporal Aggregation - method of fusing features from different frames.
- Timesteps Used - number and temporal difference of timesteps used in aggregation.
- Candidate Location Processing - the modules used for processing the multi-timestep features aggregated to candidate locations.
- Task/Supervision - the task the candidate locations are used for and the corresponding loss.

Table 1 distills works into these components, and full details can be found in Appendix A.

## 3.3 CONNECTING MULTI-VIEW STEREO AND TEMPORAL 3D DETECTION

To connect multi-view stereo to temporal 3D detection, we note the following key point: In multi-view stereo, the model predicts the likelihood that a candidate location is occupied *by anything* while temporal 3D detection models predict whether the location is occupied by *a certain object*. At their core, both methods have the same goal - given a candidate 3D location, they both consider how that location is captured in multiple 2D views to determine whether something of interest is there.

This connection holds for both grid-based and query-based methods. Grid-based methods generate *dense* candidate locations that cover the entire 3D or BEV region. Supervised by a heatmap loss Yin et al. (2021), a candidate location predicts high probability if an object exists at that location. This is similar to MVSNet which maximizes probabilities at candidate locations where anything exists. On the other hand, query-based methods use *sparse* candidate locations around where objects are likely to exist. These queries are supervised by L1 to predict offsets that move query locations towards object centers. This can be formulated as the query predicting a Laplacian distribution center in 3D space with unit scale, which is directly analogous to the predicted and iteratively refined L2-supervised Gaussian distribution over depths in MaGNet. Overall, only the specific details of the various components change between models - in essence, both temporal grid-based and temporal query-based methods aggregate or compare multi-view 2D features to make occupancy predictions, which is intrinsically analogous to multi-view stereo matching (Yang et al., 2020).

## 3.4 ANALYSIS OF COMPONENTS

Having represented 3D detection methods as instances of temporal stereo under our framework, we examine their various components to identify key points hindering their temporal fusion. From

**Optimal Time Difference for Various Depth Hypotheses and X Image Coordinates**

Figure 4: Optimal time difference over candidate locations. White regions have no valid projection.

Table 1, we observe that most 3D detection are significantly limited in the number of frames and the time window they fuse, with only BEVFormer and UniFormer considering longer fusion. However, BEVFormer demonstrates no improvement from training on more than 3 timesteps, showing that their sequential fusion framework hinders the potential for long-term temporal fusion. Further, UniFormer does not evaluate on 3D detection and their performance on map segmentation peaks with 3 seconds of history. Compared with MVS works that all use a high 1/4 resolution, many detection methods use low 1/16 resolution feature maps for object detection temporal stereo. Although some works do benefit from higher-resolution feature maps, notably MVS-inspired works such as STS, BEVStereo, and DfM, their short temporal window and few frames caused by computational limitations hinder their localization ability. We will show in Section 4 that limited temporal fusion and low resolution features in prior work are both factors that limit the localization potential of existing methods, motivating the design choices of our method that leverages *both* long-term temporal fusion and high-resolution features while maintaining high efficiency.

## 4 THEORETICAL ANALYSIS

In Section 3, we formulated temporal camera-only 3D detection works as instances of multi-view stereo. This allows us to analyze the multi-view object localization ability of these works in context of a multi-view stereo setup. In this section, we focus on the general two-view setting and perform a theoretical analysis on how realistic changes between views affect the ease of multi-view depth estimation. Specifically, we derive a formulation for our defined localization potential and examine how *temporal differences* between the views and *image resolution* affect the potential.

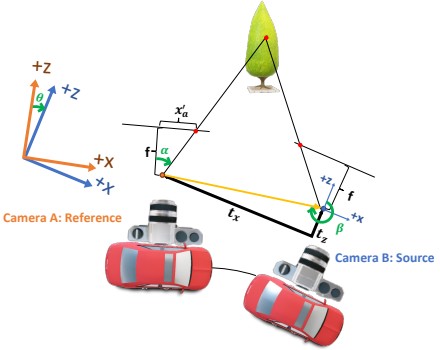

Figure 3: Reference and source coordinate systems. Arrows indicate positive angle direction.

### 4.1 DERIVATION OF LOCALIZATION POTENTIAL

Let image A be the reference view we predict depth in and let image B be the source view we leverage for multi-view depth estimation. Further let $(x_a, y_a, d_a)$ denote a pixel $(x_a, y_a)$ in image A with depth $d_a$, our candidate point, and let $(x_b, y_b, d_b)$ be its corresponding projection onto image B. Camera and view details are in Appendix B.1 and shown in Figure 3. We previously defined localization potential as the magnitude of the change in the source-view projection induced by a change in the depth in the reference view. Focusing our analysis on $x_b$ as $y_b$ is a sub-case of $x_b$ with $\theta, t_x = 0$, we represent localization potential at $(x_a, y_a, d_a)$ as:

$$\text{Localization Potential} = \left| \frac{\partial x_b}{\partial d_a} \right| = \frac{f\bar{t}\cos(\alpha)|\sin(\alpha - (\theta + \beta))|}{(d_a \cos(\alpha - \theta) + t_z \cos(\alpha))^2} \qquad (1)$$

where $\alpha, \beta, \theta$ are the angles of image A x-coordinate, translation, and rotation between views shown in Figure 3, and $\bar{t}$ is the magnitude of translation $\sqrt{t_x^2 + t_y^2}$. The full proof is in the Appendix B.

### 4.2 EFFECT OF TEMPORAL DIFFERENCE ON LOCALIZATION POTENTIAL

We consider how the temporal difference between the reference image A and the source image B effects localization potential. To do so, we re-write Equation 1 by introducing time offset $t^2$

$$\left| \frac{\partial x_b}{\partial d_a} \right| = \frac{f\bar{t}t\cos(\alpha)|\sin(\alpha - (\theta + \beta))|}{(d_a \cos(\alpha - \theta) + t_z t \cos(\alpha))^2} \qquad (2)$$

---

[2]In the main paper, we limit the effect of time to translation and exclude $\theta$ because while it is common for a vehicle to maintain the same rate of translation over long periods of time, rotation often varies from timestep to timestep as the vehicle makes small corrections. Over multiple timesteps, the aggregate $\theta$ is close to 0, and we adopt this setting in the main paper. We do, however, analyze effects of rotation on localization potential in Appendix B.4 and also examine situations such as turns where $\theta$ varies with time in Appendix B.6.

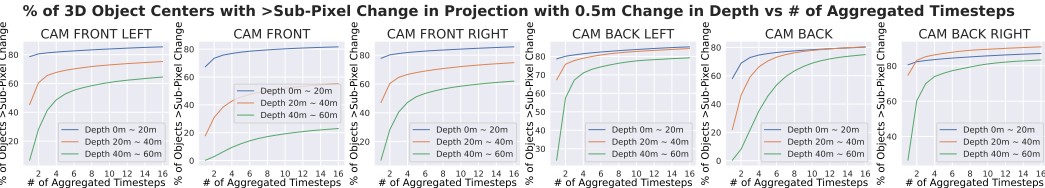

Figure 5: Visualization of relative increase in localization potential from using multiple timesteps. Note that each camera heatmap has a different scale.

Figure 6: Visualization of % of objects that can benefit from multi-view stereo.

For each candidate location over various depths and X-axis image A coordinates, we find the timestep difference between the reference and source views that maximizes its localization potential. First only allowing projections within the same camera, we observe in Figure 4 that each camera has unique trends and that closer depths prefer smaller time differences. A more extensive breakdown also considering projections between cameras and variations in $\theta$ over time is in Appendix B.5. To summarize this more complex case, we find that the optimal time difference between views varies wildly over even neighboring candidate points. As such, it is sub-optimal to determine a single time difference over which to do matching as prior works do. **Instead, leveraging many timesteps over longer history significantly increases localization potential over candidate locations**.

### 4.3 IMPACT OF RESOLUTION ON LOCALIZATION POTENTIAL

We now consider the impact of image feature resolution on localization potential. Since downsampling image features by 4x decreases the effective focal length by 4x, the localization potential is also divided by 4 from Equation 2. We consider 4 because the standard multi-view matching resolution is 1/4 and the common detection temporal stereo resolution from Section 3 is 1/16. Although deep features can maintain some intra-pixel localization information even at a lower resolution, downsampling still makes multi-view matching more difficult. As analyzed in Section 3, image feature resolution is often constrained by computational limitations. However, we find that the increased localization potential from aggregating more timesteps can compensate for the decrease in potential caused by downsampling. In Figure 5, we show the relative increase in localization potential from aggregating more timesteps and find that for most of the candidate locations, the localization potential is improved by far more than a factor of 4. Furthermore, the locations with less than a factor of 4 improvement are generally close depths for which monocular depth estimation has good performance and localization potential is already high enough. Thus, our analysis shows that **aggregating more timesteps can compensate for a lower image feature resolution, maintaining high efficiency with same localization potential as high-resolution feature matching.**

### 4.4 EFFECT OF TEMPORAL DIFFERENCE ON MULTI-VIEW DEPTH AMBIGUITY

So far, we have shown theoretically that aggregating multiple timesteps allows for more diverse rotations and greater localization potential for various candidate locations. We conclude our analysis by directly verifying the impact of aggregating multiple timesteps on multi-view depth ambiguity. More specifically, we consider the projection difference induced by varying the 3D centers of objects in the nuScenes training set by 0.5m$^3$. Using frame-to-frame ego-motion provided by nuScenes, Figure 6 shows the percent of 3D centers with a projection difference of at least one pixel, which is the minimum necessary to accurately localize a point in 3D using multiple views. A more thorough analysis is in Appendix B.7, and we conclude that **our theoretical analysis holds true empirically - with a larger number of aggregated timesteps, the percent of objects that can benefit from multi-view stereo dramatically increases over all cameras and depths.**

### 5 METHOD

We propose SOLOFusion, a natural extension of our theoretical analysis. Core to our method and design choices is balancing the impacts of image resolution and temporal aggregation on localization potential. SOLOFusion has two main streams: 1) LSS-based object detection temporal stereo for

---

[3]Most methods adopt this granularity for depth prediction (Li et al., 2022c;b; Liu et al., 2022c)

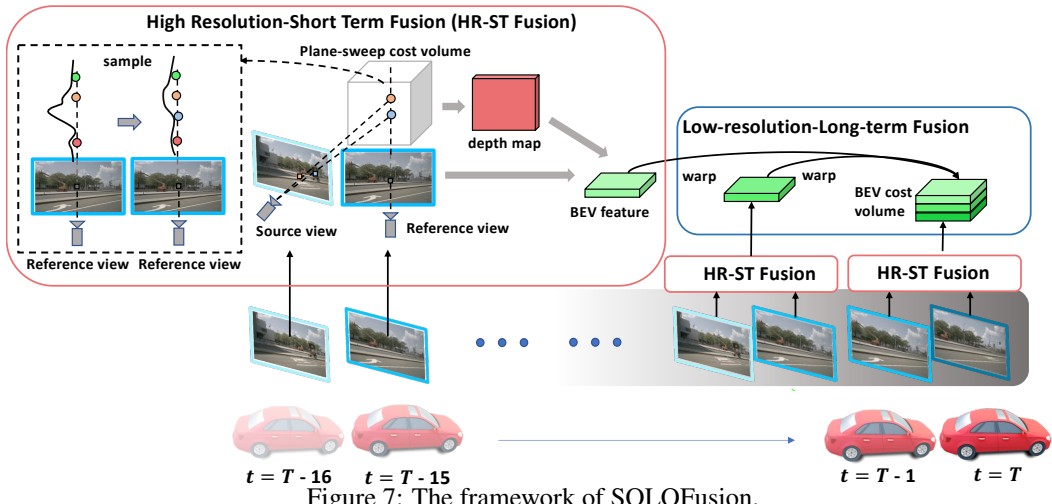

Figure 7: The framework of SOLOFusion.

coarse image features using a long history; 2) MVS-based depth prediction temporal framework for high resolution image features using a short history. The framework is presented in Figure 7.

### 5.1 LOW-RESOLUTION, LONG-TERM OBJECT DETECTION TEMPORAL STEREO

Using image features and depth prediction at a coarse 1/16 resolution, we densely generate a point cloud and voxelize into a BEV feature map similar to (Huang et al., 2021; Li et al., 2022c). Different from these methods, we offset the deterioration in localization potential caused by the low image resolution by generating a BEV cost volume leveraging long history. Specifically, we align the BEV feature maps from the previous $T$ timesteps to the current timestep and concatenate them. Then, we use the resulting long-term BEV cost volume for detection. We find that this straightforward and efficient pipeline already outperforms existing methods by a large margin.

### 5.2 HIGH-RESOLUTION, SHORT-TERM DEPTH ESTIMATION TEMPORAL STEREO

We then consider using high-resolution image features to offset short temporal history. To improve the per-frame depth estimation used for object detection temporal stereo in Section 5.1, SOLOFusion leverages an MVS temporal stereo pipeline using two consecutive frames. We consider a plane-sweep volume in the reference view at 1/4 resolution and perform stereo matching with the previous frame image features. Opting for a straightforward pipeline to test our analysis, we simply add the cross-view correlations with the monocular depth predictions, supervising them together.

As this naive setup is computationally expensive, we propose to use monocular depth prediction to guide the stereo matching. we draw inspiration from the exploration and exploitation trade-off common in reinforcement learning. Stereo matching a candidate depth is expensive and is analogous to collecting an observation in RL. We hypothesize that balancing exploitation, using the monocular prior, and exploration, considering other candidate locations for stereo, when generating depth hypotheses can best leverage the expensive temporal matching. Hence, we propose Gaussian-Spaced Top-k sampling, an iterative framework to generate sampling points. We first choose the depth hypothesis with the highest monocular probability, exploiting our prior. Then, to force exploration, we down-weight the monocular probability of its neighbors, forcing exploration beyond simple prior maxima. We repeat this process, yielding a set of $k$ candidate locations for each pixel, and perform temporal stereo on these points. Additional details can be found in Appendix C.

We find that this pipeline is able to cover multi-modal depth distributions and far outperforms simply top-k sampling with minimal computational cost. Our high-resolution, short-term depth estimation module further improves on the strong low-resolution, long-term temporal stereo model, demonstrating that short and long-term temporal stereo are complementary.

*Necessity of Balance.* Localization potential is maximized if we use high-resolution, long-term fusion. However, we find that even high-resolution, short-term temporal stereo imposes significant memory and runtime costs without our proposed matching point sampling method, making extension to high-resolution, long-term intractable. Our pipeline is built on our theoretical analysis to find a good trade-off between resolution and time difference, using improvements in one to compensate for deterioration in the other, to maximize localization potential under computational constraints.

Table 2: Comparison on the nuScenes `val` set. Note that in each category, we evaluate against methods with a comparable or stronger setting than us. [†]Initialized from FCOS3D backbone.

| Methods | Backbone | Image Size | CBGS | mAP↑ | NDS↑ | mATE↓ | mASE↓ | mAOE↓ | mAVE↓ | mAAE↓ |
|---|---|---|---|---|---|---|---|---|---|---|
| BEVDet | ResNet50 | 256 × 704 | ✓ | 0.298 | 0.379 | 0.725 | 0.279 | 0.589 | 0.860 | 0.245 |
| PETR | ResNet50 | 384 × 1056 | ✓ | 0.313 | 0.381 | 0.768 | 0.278 | 0.564 | 0.923 | 0.225 |
| BEVDet4D | ResNet50 | 256 × 704 | ✓ | 0.322 | 0.457 | 0.703 | 0.278 | 0.495 | 0.354 | 0.206 |
| BEVDepth | ResNet50 | 256 × 704 | ✓ | 0.351 | 0.475 | 0.639 | **0.267** | 0.479 | 0.428 | 0.198 |
| STS | ResNet50 | 256 × 704 | ✓ | 0.377 | 0.489 | 0.601 | 0.275 | 0.450 | 0.446 | 0.212 |
| BEVStereo | ResNet50 | 256 × 704 | ✓ | 0.372 | 0.500 | 0.598 | 0.270 | **0.438** | 0.367 | 0.190 |
| SOLOFusion | ResNet50 | 256 × 704 | ✓ | **0.427** | **0.534** | **0.567** | 0.274 | 0.511 | **0.252** | **0.188** |
| FCOS3D | ResNet101-DCN | 900 × 1600 | ✗ | 0.295 | 0.372 | 0.806 | 0.268 | 0.511 | 1.131 | 0.170 |
| BEVFormer[†] | ResNet101-DCN | 900 × 1600 | ✗ | 0.416 | 0.517 | 0.673 | 0.274 | 0.372 | 0.394 | 0.198 |
| PolarDETR-T[†] | ResNet101-DCN | 900 × 1600 | ✗ | 0.383 | 0.488 | 0.707 | 0.269 | **0.344** | 0.518 | **0.196** |
| UVTR[†] | ResNet101-DCN | 900 × 1600 | ✗ | 0.379 | 0.483 | 0.731 | **0.267** | 0.350 | 0.510 | 0.200 |
| PolarFormer[†] | ResNet101-DCN | 900 × 1600 | ✗ | 0.432 | 0.528 | 0.648 | 0.270 | 0.348 | 0.409 | 0.201 |
| SOLOFusion | ResNet101 | 512 × 1408 | ✗ | **0.472** | **0.544** | **0.518** | 0.275 | 0.604 | **0.310** | 0.210 |
| DETR3D[†] | ResNet101-DCN | 900 × 1600 | ✓ | 0.349 | 0.434 | 0.716 | 0.268 | 0.379 | 0.842 | 0.200 |
| PETR | ResNet101 | 512 × 1408 | ✓ | 0.357 | 0.421 | 0.710 | 0.270 | 0.490 | 0.885 | 0.224 |
| BEVDepth | ResNet101 | 512 × 1408 | ✓ | 0.412 | 0.535 | 0.565 | 0.266 | 0.358 | 0.331 | **0.190** |
| STS | ResNet101 | 512 × 1408 | ✓ | 0.431 | 0.542 | 0.525 | **0.262** | **0.380** | 0.369 | 0.204 |
| SOLOFusion | ResNet101 | 512 × 1408 | ✓ | **0.483** | **0.582** | **0.503** | 0.264 | 0.381 | **0.246** | 0.207 |

# 6 EXPERIMENTS

## 6.1 DATASET AND METRICS

We use the large scale nuScenes dataset Caesar et al. (2020), containing 750, 150, and 150 scenes for training, validation, and testing, respectively. Each sequence is 20s long and is captured using 6 cameras with resolution 900 × 1600. 3D box annotations are provided at key frames every 0.5s. We refer readers to the official paper for metrics. The full experimental details are in Appendix D.

## 6.2 MAIN RESULTS

We report our results on the nuScenes `val` and `test` sets in Table 2 and Appendix E. Despite often operating under weaker settings, SOLOFusion far outperforms prior state-of-the-art under every setup. Notably, in the ResNet101, CBGS setting, our framework outperforms STS, which extends BEVDepth to incorporate multi-view stereo, by a substantial 5.2% mAP. This improvement, coupled with an increase in mATE which is also a object localization metric, verifies our theoretical analysis that longer temporal fusion can indeed allow for better multi-view localization. Our method is also much better at predicting velocity, demonstrating a 12.3% improvement in mAVE, which shows that long-term temporal fusion is not just better for localization - it also improves velocity estimation by observing dynamic objects for longer. SOLOFusion obtain 1st place on the nuScenes `test` set camera-only track at time of submission. Our framework achieves this by training without CBGS, with a shorter cycle, and without large-scale depth training or test-time augmentation, demonstrating the importance of using both short-term and long-term temporal fusion.

## 6.3 ABLATION STUDY & ANALYSIS

**Ablation of Time Window for Temporal Fusion.**

We start with a BEVDepth baseline with no temporal fusion and ablate the addition of low-resolution, long-term fusion in Table 3. Although fusing a single timestep improves velocity prediction which improves NDS, the localization metrics mAP and mATE improve only slightly. As we fuse more timesteps into our

Table 3: Ablation for low-res, long-term fusion.

| # of Prev. Timesteps | mAP↑ | NDS↑ | mATE↓ | mAVE↓ |
|---|---|---|---|---|
| 0 | 0.307 | 0.347 | 0.743 | 1.148 |
| 1 | 0.316 | 0.423 | 0.734 | 0.456 |
| 2 | 0.326 | 0.434 | 0.736 | 0.382 |
| 4 | 0.349 | 0.452 | 0.701 | 0.332 |
| 8 | 0.366 | 0.465 | 0.686 | 0.317 |
| 16 | 0.377 | 0.474 | 0.655 | 0.307 |
| 41 | 0.367 | 0.467 | 0.650 | 0.314 |

BEV cost volume, the mAP and mATE increases, improving by 6.1% and and 7.9% from 1 to 16 timesteps. This supports our analysis that compared to using a single timestep, leveraging many timesteps over a larger time window substantially increases localization potential. We find that performance saturates at 16 timesteps as there is little overlap in the visible region beyond 16 timesteps. Further analysis of the impact of long-term fusion on static and moving objects are in Appendix G.

**Ablation of Depth Hypothesis Sampling.** In Table 4 we ablate various methods for choosing the depth hypotheses for high-resolution, short-term temporal stereo. Starting with the single-frame baseline, we find that attempting to match all depth hypotheses (112 for each pixel) increases runtime by 6x and significantly increases GPU memory usage. The training time and memory cost increase is even more significant, and we find it impossible to train this model properly. A model with 28 uniformly sampled hypotheses is able to improve performance, but it imposes a 2.4x slowdown

Table 4: Ablation of depth hypothesis sampling methods

| Method of Depth Sampling | # of Depth Hypotheses | FPS | Memory | mAP↑ | NDS↑ | mATE↓ |
|---|---|---|---|---|---|---|
| None; Single-Frame BEVDepth | - | 17.6 | 3.3 GB | 0.321 | 0.349 | 0.722 |
| Uniform Sampling | 112 (all) | 2.9 | 8.5 GB | - | - | - |
| Uniform Sampling | 28 | 7.4 | 4.2 GB | 0.345 | 0.377 | 0.692 |
| Uniform Sampling | 7 | 12.0 | 3.3 GB | 0.319 | 0.359 | 0.727 |
| Top-k Sampling | 7 | 11.9 | 3.3 GB | 0.336 | 0.390 | 0.674 |
| Gaussian-Spaced Top-k Sampling | 7 | 11.4 | 3.3 GB | 0.343 | 0.389 | 0.670 |

Table 5: Ablation of short-term and long-term fusion components

| | High-Res, Short-Term | Low-Res, Long-Term | FPS | Memory | mAP↑ | NDS↑ | mATE↓ |
|---|---|---|---|---|---|---|---|
| (a) | ✗ | ✗ | 17.6 | 3.3GB | 0.321 | 0.349 | 0.722 |
| (b) | ✓ | ✗ | 12.2 | 3.3GB | 0.343 | 0.389 | 0.670 |
| (c) | ✗ | ✓ | 15.9 | 3.6GB | 0.386 | 0.479 | 0.650 |
| (d) | ✓ | ✓ | 11.4 | 3.6GB | 0.404 | 0.495 | 0.605 |

Table 6: Analysis of whether many-frame temporal fusion can compensate for low-resolution.

| Method | Backbone | Image Size | FPS | Memory | mAP↑ | NDS↑ | mATE↓ |
|---|---|---|---|---|---|---|---|
| BEVDepth | ResNet50 | $512 \times 1408$ | 2.3 | 7.3 GB | 0.405 | 0.523 | 0.570 |
| SOLOFusion | ResNet50 | $256 \times 704$ | 11.4 | 3.6 GB | 0.427 | 0.534 | 0.567 |

and an increase in GPU memory. Although uniformly sampling 7 candidates improves runtime, it demonstrates no improvement from stereo. Naively leveraging the monocular depth prior with top-k sampling, we do see improvement in all metrics. Interestingly, we find that top-k sampling performs even better than the 28 uniformly sampled hypotheses for localization (mATE), demonstrating that monocular priors are useful for guiding stereo matching. Replacing naive top-k with our proposed Gaussian-Spaced Top-k Sampling improves mAP and mATE with a minimal decrease in FPS.

**Ablation of High-Res, Short-Term Fusion and Low-Res, Long-Term Fusion.** Starting from a baseline BEVDepth model, we ablate the addition of our short-term and long-term fusion components in Table 5. We find that although the addition of the high-res, short-term matching from (a) to (b) slows down runtime from 17.6 to 12.2 FPS, there no increase in GPU memory cost and the localization mATE is significantly improved by 5.2%. Instead adding the low-res, long-term matching from (a) to (c), the improvement in mAP is huge with a minimal drop in FPS. We note that purely considering localization improvement, adding high-res, short-term improved by mATE by 5.2% while low-res, long-term improved by 7.2%. The comparatively similar increase shows that both modules have a similar amount of localization potential, illustrating the trade-off we analyzed in Section 4.3. Finally, adding both modules further improves performance for all metrics. We observe that the the individual improvements from the two modules, ∼5% and ∼7%, yield a combined improvement of ∼12%, demonstrating that the two sides of the trade-off are highly complementary.

**Balancing Temporal Fusion and Resolution.** In Section 4.3, we found that extensive temporal fusion can compensate for lower resolution in terms of localization potential. We verify this in Table 6 by comparing half-resolution SOLOFusion, designed to offset low resolution with long-term temporal fusion, with full-resolution BEVDepth. SOLOFusion outperforms BEVDepth by 2.2% mAP and 1.1% NDS. Furthermore, SOLOFusion demonstrates comparable object localization (mATE) despite operating at half resolution, verifying our theoretical analysis. This also demonstrates an important property of SOLOFusion. By compensating for a decrease in resolution, which dramatically decreases inference cost, with an increase in temporal fusion, which has minimal impact on latency, low-resolution SOLOFusion achieves similar performance as high-resolution BEVDepth at roughly 5x the FPS and half the memory cost. Additional runtime analysis is in Appendix F.

## 7 CONCLUSION

In this paper, we provide new outlooks and a baseline for temporal 3D object detection. We reformulate the multi-view 3D object detection as an instance of temporal stereo matching and define *localization potential* to measure the ease of multi-view depth estimation. Through detailed theoretical and empirical analyses, we demonstrate the importance of temporal difference and granularity of features for temporal stereo. We find that using long temporal history significantly increases localization potential even with low-resolution features. Our analyses lead to a natural baseline, termed SOLOFusion, which takes advantage of the synergy of short and long-term temporal information as well as the trade-off between time and resolution. Extensive experiments demonstrate the effectiveness of the proposed method and SOLOFusion achieves a new state-of-the-art on the nuScenes dataset, outperforming prior-arts by 5.2% mAP and 3.7% NDS and placing first in the leaderboard.

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

# A  ADDITIONAL DETAILS FOR THE UNIFIED TEMPORAL STEREO FORMULATION FOR CAMERA-ONLY 3D DETECTION

In this section, we explain how the various temporal methods can be organized into our framework in Section 3. We group methods by their type.

## A.1  MULTI-VIEW STEREO METHODS

**MVSNet.** As the seminal multi-view stereo work, MVSNet fits directly into our framework. Its candidate locations considered for matching is every $(x, y, d)$ point in the reference camera view. Image features are obtained for these points via simple projection to each view at a fine 1/4 resolution and subsequent bilinear sampling. Having obtained image features from multiple views - two source views and one reference view - for each candidate location, these features are aggregated through a variance metric. Finally, the candidate locations in the plane-sweep volume are processed using 3D convolutions, and MVSNet predicts a probability for every $(x, y, d)$ candidate location using the aggregated temporal information. Intuitively, this probability value is a prediction of *whether the candidate location is occupied in 3D space*. Finally, this probability distribution over depths for every pixel is used to take a weighted sum over depth locations, yielding a single depth prediction for every pixel, which is then supervised by L1 loss. Our formulation is able to encapsulate every component of a representative multi-view/temporal stereo framework, and we find it generalizes naturally to temporal 3D detection.

**MaGNet.** As another MVS work, MaGNet can also be deconstructed into our components. Its candidate locations are $(x, y, d)$ points within the confidence interval of the predicted depth Gaussian. The rest of the pipeline is similar to that of MVSNet, only different in that it uses dot product instead of variance to aggregate temporal features and uses a 2D CNN to process the temporal features at the candidate locations. Using these processed features, MaGNet iteratively updates the depth Gaussian and is supervised via L2 to align the Gaussian with the true ground truth depth.

## A.2  GRID-BASED 3D DETECTION METHODS

**BEVDet4D.** The candidate locations are BEV grid cells, and each cell samples features by pooling over $(x, y, d)$ image points that fall within each cell. As pooling is done over densely generated depth hypotheses, the out-projected image feature resolution is kept coarse at 1/16. BEVDet4D aligns a previous BEV feature map with the current one using ego-motion and simply concatenates them. Considering this alignment in reverse, a grid location receives features from $(x, y, d)$ past image points that were within that grid cell location before. In this manner, each grid cell receives image features of multiple timesteps from regions in 2D views around the grid cell's 2D projection. For inference, they use a single historical observation at 2.5s and process the fused BEV map with 2D convolutions for object prediction.

**BEVDepth.** BEVDepth, as an extension of BEVDet, maintains the exact same main components as BEVDet4D. It does, however, use a more recent frame of 0.5s for aggregation.

**STS.** Already briefly discussed in the main paper, STS extends the BEVDepth pipeline, and hence the BEVDet4D pipeline, by adding an MVS depth estimation temporal stereo component. The components that make up the MVS depth estimation component of STS are mostly identical to that of MVSNet. Although STS appears to leverage both low and high resolution temporal stereo, their low-resolution temporal stereo inherited from BEVDepth is still severely hindered by the single temporal frame they use, limiting the improvements STS can obtain from utilizing temporal information.

**BEVStereo.** Similar to STS, BEVStereo extends the BEVDepth pipeline by adding an MVS depth estimation temporal stereo module. However, instead of drawing from MVSNet, BEVStereo instead adopts intuitions from MaGNet.

**DfM.** Similar to STS and BEVStereo, DfM is an MVS-inspired detection method. However, instead of group-wise correlation, DfM uses concatenation of temporal features. Further, they use 2D & 3D convolutions to process the aggregated features before using them for both depth estimation and object prediction.

**MV-FCOS3D++.** For candidate locations, they use 3D voxel centers, in LiDAR coordinates, and they project these voxels to the image plane to sample features. This is unlike LSS-based works that generate a dense point cloud from image features and use voxelization steps. Similar to DfM, they concatenate multiple timesteps' image features in the 3D volume and process the 3D volume with 3D and 2D convolutions. Unlike DfM, they do not do depth estimation, only doing object prediction.

## A.3 QUERY-BASED 3D DETECTION METHODS

**BEVFormer.** The candidate locations are queries aligned to BEV cell locations, and each query has fixed 3D sampling locations along the z-axis which are projected onto images. BEVFormer uses several coarse image feature maps from 1/16 to 1/64 resolution. The previous timestep BEV queries are aligned to the current frame using ego-motion their features are aggregated to current queries using deformable attention. Similar to the formulation of BEVDet4D, this temporal alignment allows current BEV query candidate locations to aggregate image features that correspond to it in a previous timestep. We note that BEVFormer continuously saves the past BEV queries for the next timestep, allowing for continuous temporal fusion. However, they observe no benefit beyond training using three historical timesteps, limiting their potential for long-term temporal fusion. Finally, a transformer decoder is used on the BEV queries for object prediction.

**UniFormer.** UniFormer adopts the same BEV queries as BEVFormer, but uses sampling isntead of deformable attention to aggregate image features. UniFormer uses 1/8 to 1/64 resolutions and aggregates previous timesteps' information by projecting sampling locations to previous timestep images and fusing multi-timestep features through learned weighted sum. This work experiments with longer time history, up to 10 past timesteps, but they observe no gains beyond 6 timesteps (3 seconds), and they adopt this setting for their final model. UniFormer is supervised with map segmentation only.

**PolarDETR.** Although similar to BEVFormer, PolarDETR instead uses moveable object queries as candidate sampling locations. Projecting the predicted object query locations on to images, using temporal ego-motion transformation to previous timesteps, and aggregating features using deformable attention, PolarDETR combines features from different timesteps though concatenation. PolarDETR only utilizes a single frame at 0.5s history and processes the object queries with a transformer decoder for object prediction. As explained in Section 3.3 in the main paper, the supervised offsets of the object query location that moves it towards where objects are likely to exist causes PolarDETR to be a sparse candidate location version of multi-view stereo, localizing object occupancy regions in 3D space using temporal information.

**PolarFormer.** Like other query-based methods, the candidate locations are object queries. However, different from other works that directly aggregate image features to to object queries, PolarFormer generates an intermediate Polar BEV representation that image features are first projected to. The candidate locations then aggregate features from this Polar BEV feature map. Despite this intermediate step, the key points are still similar to other query-based works. As the object queries perform deformable attention on the Polar BEV representation, there is still a significant prior that forces queries to aggregate features from BEV locations close to it. Since this Polar BEV representation was generated using projection and alignment of image locations, there is still a strong spatial connection between regions where the object query's aggregated features come from and the object query's projection onto the image. Unfortunately, the paper does not specify the resolution of image features used. The object query also has access to multiple views of it over time as previous Polar BEV representations are aligned and concatenated. The remainder of the pipeline is standard for query-based methods, with a transformer decoder then predicting objects from the candidate locations.

**UVTR.** UVTR is similar to PolarFormer in its use of an intermediate representation. Instead of a Polar BEV feature map, UVTR generates a 3D volume whose voxels are projected onto images to get their timesteps. Past 3D volumes are aligned and concatenated, and object queries, which are the candidate locations, aggregate image features through deformable attention onto this 3D volume, allowing our formulation for PolarFormer to be directly applied to UVTR.

**PETRv2.** PETRv2 is different from previous works in that candidate locations aggregate image features not through projection or deformable attention but through unconstrained cross-attention over both previous and current image features. However, by out-projecting image features and

decorating them with 3D positional emebddings, PETRv2 encourages object queries to attend to image features that are spatially relevant to it. Furthermore, PETR shows that object queries attend most to image locations that they are projected to, making this cross-attention operation a form of "soft" projection and sampling. From this observation, the rest of the analysis is identical to that of PolarDETR.

## B  ADDITIONAL DETAILS FOR THEORETICAL ANALYSIS

### B.1  DETAILS OF REFERENCE AND SOURCE CAMERA SETUP

We define the camera intrinsics, inverse intrinsics, and camera A to camera B transformation as:

$$
K = \begin{bmatrix} f & 0 & c_x \\ 0 & f & c_y \\ 0 & 0 & 1 \end{bmatrix}, K^{-1} = \begin{bmatrix} 1/f & 0 & -c_x/f \\ 0 & 1/f & -c_y/f \\ 0 & 0 & 1 \end{bmatrix}, Rt_{A\to B} = \left[ \begin{array}{ccc|c} \cos\theta & 0 & -\sin\theta & t_x \\ 0 & 1 & 0 & 0 \\ \sin\theta & 0 & \cos\theta & t_z \end{array} \right] \tag{3}
$$

where $\theta$ is the rotation from camera A to camera B in the XZ plane and $t_x$, $t_z$ are translations from camera A to B as shown in Figure 3. We exclude transformations involving the vertical Y axis as they are minimal in driving scenarios. Then, the projection of $(x_a, y_a, d_a)$ onto image B is:

$$
[x_b, y_b] = \left[ \frac{d_a x'_a \cos\theta - d_a f \sin\theta + t_x f}{\frac{d_a x'_a \sin\theta}{f} + d_a \cos\theta + t_z} + c_x, \frac{d_a y'_a}{\frac{d_a x'_a \sin\theta}{f} + d_a \cos\theta + t_z} + c_y \right] \tag{4}
$$

where $x'_a = x_a - c_x, y'_a = y_a - c_y$.

### B.2  DERIVATION AND ANALYSIS OF IMAGE A TO IMAGE B PROJECTION

In this section, we derive our formulation for the projection $(x_b, y_b, d_b)$ in Equation 4 and connect it to the familiar standard stereo case.

Starting from Image A coordinates $(x_a, y_a, d_a)$ and applying homographic transforms:

$$
\begin{aligned}
[x_a, y_a, d_a] &\implies [x_a d_a, y_a d_a, d_a] && \text{(Image A Image Hom Coords)} \\
&\implies \left[ \frac{d_a(x_a - c_x)}{f}, \frac{d_a(y_a - c_y)}{f}, d_a \right] && \text{(Image A Camera Coords)} \\
&\implies \left[ \frac{d_a x'_a}{f}, \frac{d_a y'_a}{f}, d_a \right] && \text{(Let } x'_a = x_a - c_x, y'_a = y_a - c_y) \\
&\implies \left[ \frac{d_a x'_a \cos\theta}{f} - d_a \sin\theta + t_x, \frac{d_a y'_a}{f}, \frac{d_a x'_a \sin\theta}{f} + d_a \cos\theta + t_z \right] \\
&&& \text{(Image B Camera Coords)}
\end{aligned}
$$

$$
\begin{aligned}
&\implies \begin{bmatrix} \dfrac{d_a x'_a \cos\theta - d_a f \sin\theta + t_x f}{\frac{d_a x'_a \sin\theta}{f} + d_a \cos\theta + t_z} + c_x, \\[2ex] \dfrac{d_a y'_a}{\frac{d_a x'_a \sin\theta}{f} + d_a \cos\theta + t_z} + c_y, \\[2ex] \dfrac{d_a x'_a \sin\theta}{f} + d_a \cos\theta + t_z \end{bmatrix} && \text{(Image B Image Coords)} \\[2ex]
&= [x_b, y_b, d_b] && \text{(Image B Image Coords)}
\end{aligned}
$$

This formulation is applicable to any two-camera system with rotation and translation along the XZ axis. For instance, consider a standard stereo setup with cameras $A$ and $B$ as the left and right cameras, respectively: $\theta = 0$, $t_z = 0$, baseline $-t_x > 0$. Then, the above reduces to:

$$
[x_b, y_b, d_b] = \left[ \frac{d_a x'_a + t_x f}{d_a} + c_x, \frac{d_a y'_a}{d_a} + c_y, d_a \right] = \left[ x_a - \frac{t_x f}{d_a}, y_a, d_a \right]
$$

This yields the standard stereo disparity formula $disparity = \frac{t_x f}{d_a}$. For depth estimation in stereo or temporal stereo, we project multiple depth hypotheses for a pixel in image A onto image B and find the pixel along the epipolar line in image B that matches best with the original pixel in image A. Given such a matching, we can derive the depth using the transformation matrix between the two

images/cameras. For such a formulation to work well, it is beneficial for the image B projections of nearby depth hypotheses for a pixel in image A to be as far apart from one another as possible. For instance, if two candidate depths are projected to the same pixel in image B, it is impossible to determine which of the two candidates are a better match. Even beyond same-pixel projections, with downsampled feature maps and local homogeneity of features extracted from CNN backbones, more separated depth hypothesis projections allows for easier stereo depth estimation. To quantify changes in projection from changes in depth, we defined in the main paper localization potential, which is closely tied to the ease of depth estimation.

We then examine localization potential in this standard stereo case by finding $|\frac{\partial x_b}{\partial d_a}|$. For the standard stereo setup, $|\frac{\partial x_b}{\partial d_a}| = \frac{(-t_x)f}{d_a^2}$. This partial derivative tells us that localization potential is larger, which means depth estimation is easier, if:

- The baseline $-t_x$ is larger. This is in-line with our intuition; the further apart the cameras are, smaller depth changes can be captured.

- The focal length $f$ is larger. Intuitively, if we downsample the image resolution, the focal length decreases, causing more different depth hypotheses to project to the now "larger" pixels.

- The depth at which we evaluate localization potential is smaller. Indeed, the projected difference between 1m and 2m is larger than the difference between 59m and 60m.

We do comment, however, that these observations do not mean the depth estimation quality in standard stereo can be simply improved by adopting a larger baseline and focal length. A larger baseline significantly decreases the overlapping region in standard stereo while a larger focal length limits the scene captured. In addition, we observe that unlike the general two-view case in Equation 1, the localization potential does not vary over different image A x-coordinates for standard stereo. This is because the stereo cameras are aligned, causing the epipolar lines to be parallel to the x axis. As such, choosing an optimal setup for standard stereo is much simpler compared to the more general multi-view, temporal stereo case.

### B.3 Full Proof of Formulation of Localization Potential

In this section, we detail the steps we took to derive our formulation for localization potential in Equation 1. We first reparameterize $x_b$ using $\alpha$ as defined in Figure 3. Let $r'_a = \sqrt{(x'_a)^2 + f^2}$ and note that we have $\sin\alpha = \frac{x'_a}{r'_a}, \cos\alpha = \frac{f}{r'_a}$.

$$
\begin{aligned}
x_b &= \frac{d_a x'_a \cos\theta - d_a f \sin\theta + t_x f}{\frac{d_a x'_a \sin\theta}{f} + d_a \cos\theta + t_z} + c_x \\[2ex]
&= f \frac{d_a x'_a \cos\theta - d_a f \sin\theta + t_x f}{d_a x'_a \sin\theta + d_a f \cos\theta + t_z f} + c_x && \text{(multiply top \& bottom with } f) \\[2ex]
&= f \frac{d_a r'_a \left(\frac{x'_a}{r'_a}\cos\theta - \frac{f}{r'_a}\sin\theta\right) + t_x f}{d_a r'_a \left(\frac{x'_a}{r'_a}\sin\theta + \frac{f}{r'_a}\cos\theta\right) + t_z f} + c_x \\[2ex]
&= f \frac{d_a r'_a \left(\sin\alpha\cos\theta - \cos\alpha\sin\theta\right) + t_x f}{d_a r'_a \left(\sin\alpha\sin\theta + \cos\alpha\cos\theta\right) + t_z f} + c_x && \text{(substitute } \sin\alpha, \cos\alpha) \\[2ex]
&= f \frac{d_a r'_a \sin(\alpha-\theta) + t_x f}{d_a r'_a \cos(\alpha-\theta) + t_z f} + c_x && \text{(sin \& cos identities)} \\[2ex]
&= f \frac{d_a \sin(\alpha-\theta) + t_x \frac{f}{r'_a}}{d_a \cos(\alpha-\theta) + t_z \frac{f}{r'_a}} + c_x && \text{(divide top \& bottom with } r'_a) \\[2ex]
&= \frac{d_a f \sin(\alpha-\theta) + t_x f \cos\alpha}{d_a \cos(\alpha-\theta) + t_z \cos\alpha} + c_x && \text{(substitute } \cos\alpha)
\end{aligned}
$$

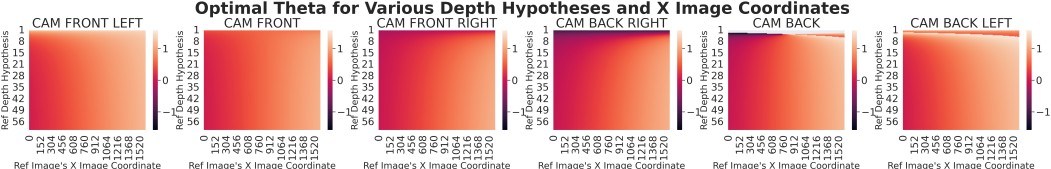

Figure 8: We visualize the optimal theta for various candidate locations.

Then, deriving $\frac{\partial x_b}{\partial d_a}$ from this formulation, we get:

$$\left|\frac{\partial x_b}{\partial d_a}\right| = \frac{f\cos(\alpha)|t_z\sin(\alpha-\theta) - t_x\cos(\alpha-\theta)|}{(d_a\cos(\alpha-\theta) + t_z\cos(\alpha))^2}$$

Note that $\cos(\alpha) > 0$ for all values of $\alpha$. We can then rewrite the above further by using the angular direction of translation $\beta$ as defined in Figure 3. The total magnitude of translation is $\bar{t} = \sqrt{t_x^2 + t_y^2}$, giving us $\sin\beta = t_x/\bar{t}, \cos\beta = t_z/\bar{t}$. Then, we get:

$$\left|\frac{\partial x_b}{\partial d_a}\right| = \frac{f\cos(\alpha)|t_z\sin(\alpha-\theta) - t_x\cos(\alpha-\theta)|}{(d_a\cos(\alpha-\theta) + t_z\cos(\alpha))^2}$$
$$= \frac{f\bar{t}\cos(\alpha)|\cos(\beta)\sin(\alpha-\theta) - \sin(\beta)\cos(\alpha-\theta)|}{(d_a\cos(\alpha-\theta) + t_z\cos(\alpha))^2}$$
$$= \frac{f\bar{t}\cos(\alpha)|\sin(\alpha-(\theta+\beta))|}{(d_a\cos(\alpha-\theta) + t_z\cos(\alpha))^2}$$

which is our formulation in Equation 1.

### B.4 EFFECT OF VIEW ROTATION $\theta$ ON LOCALIZATION POTENTIAL

In this section, we identify the impact of $\theta$ on localization potential formulation in Equation 1. Again, recall that a larger value of $\left|\frac{\partial x_b}{\partial d_a}\right|$ means easier depth estimation. *Critically*, we first observe that $\theta$ only affects the ease of depth estimation of a pixel through its *relative* angular difference with $\alpha$ of that pixel and $\beta$. This is important because it means there is no singular camera rotation that is best for all pixels or all camera translations. The optimal rotation maximizing localization potential changes based on the pixel and the current ego-motion.

Analyzing Equation 1, we find that if $\theta$ is close to $\alpha$, the $d_a\cos(\alpha-\theta)$ term increases, making the depth estimation more difficult by a factor of $d_a^2$. This is in-line with our intuition; if the camera rotation is such that the resulting camera B's principal axis is in-line with the pixel ray, the depth hypotheses along that pixel ray be projected close together. Further, we also want $\theta + \beta$ to be different from $\alpha$ (otherwise the numerator decreases). This follows a similar intuition as before - for depth hypotheses along a pixel ray to be projected further apart, the ego-vehicle should not rotate towards or move in the same direction as the pixel ray.

We empirically verify our analysis by visualizing the optimal $\theta$ that maximizes localization potential over various depths and X-axis Image A Coordinates for the six cameras in nuScenes. For translational movement, we take the average ego-motion for moving scenes in nuScenes, which yields approximately $tx = 0.05m, tz = 3.19m$ between consecutive frames (0.5s difference) in the front camera coordinates. The results are shown in Figure 8. Indeed, we find that **the optimal $\theta$ various over different pixel locations, depths, and cameras**, varying most significantly over image location. As the translation direction $\beta$ is different for each camera, each with its own coordinate system, by observing varied rotation values over different cameras, we verify that translation direction significantly affects the optimal $\theta$ as well. Examining the front camera with translation direction $\beta$ close to 0, we find that the $\theta$ seeks to maximize the difference between $\alpha - \theta$ while keeping the candidate point in-view. Furthermore, the optimal $\theta$ does change over depth as well. For instance, the optimal $\theta$ along the center ray in the back right camera changes from 0 degrees at 10m to 30 degrees at 50m. That some **candidate locations prefer smaller rotations** runs contrary to methods used to choose matching frames in indoor temporal stereo, which impose a minimum rotation and translation (Hou et al., 2019; Sun et al., 2021) between frames to be used to for matching. Hence,

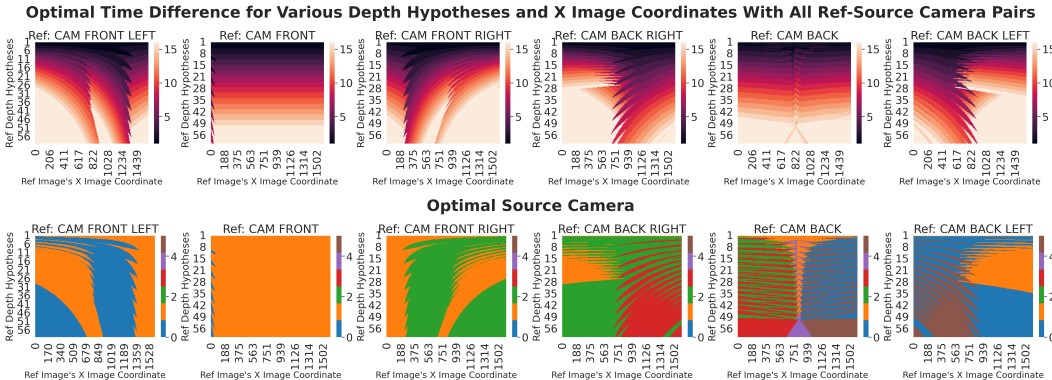

Figure 9: Allowing projections of depth hypotheses between different cameras, we visualize the optimal time difference in the first row. The second row shows the source camera that the depth hypothesis was projected onto to maximize localization potential. The cameras are labeled 0 to 5 in the order of front-left to back-left.

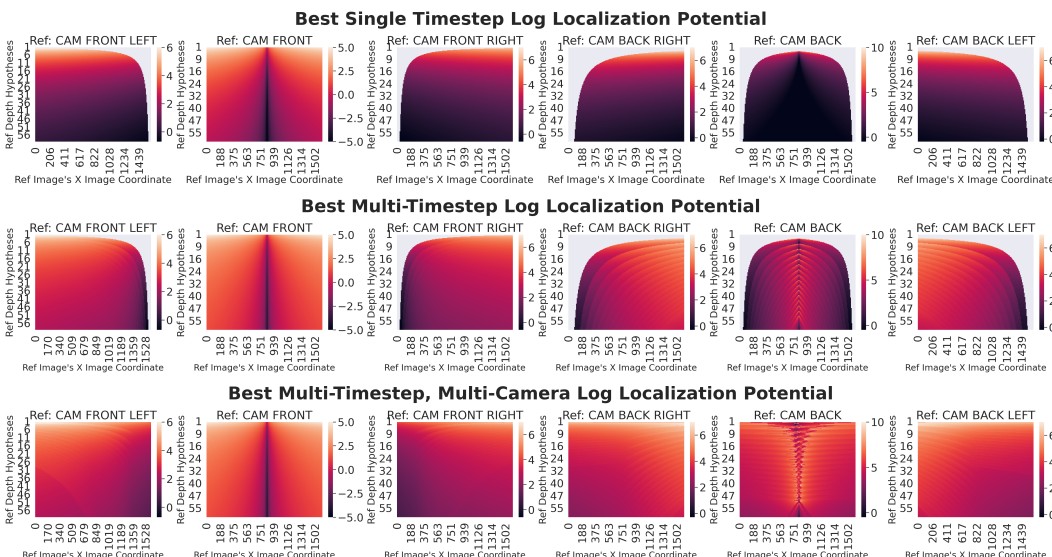

Figure 10: Log of maximized localization potential values are visualized for single-timestep with same-camera projections in row 1. Row 2 shows multi-timestep same camera projections, and row 3 shows multi-timestep, multi-camera projections. Note each vertical column has same scale

there is no globally optimal rotation between views. **To allow different candidate locations to maximize their localization potential, it is important to utilize many views with different rotations irrespective of their magnitude.** In practice, we can obtain diverse rotations by utilizing many timesteps over long history.

## B.5 OPTIMAL TIME DIFFERENCE CONSIDERING MULTI-CAMERA PROJECTION

We first further analyze trends in optimal time difference in 4 Intuitively, for the forward-facing cameras where a larger time difference increases the distance between a 3D point and the vehicle, this is a trade-off between the tendency of further depth points to be projected closer together (the denominator) and the larger difference in views generated through ego-motion (the numerator). The former wins out for closer points and the latter for further points. The different trends over various cameras are representative of both their orientation w.r.t ego-vehicle movement (we see tilted trends for the left/right slanted cameras) as well as their general forward/backward facing orientation. Next, we visualize the optimal time difference for candidate locations when allowing for projections different cameras. We also visualize the optimal source camera that the depth hypothesis is projected onto. The results are in Figure 9. Further, we also maximized log localization potential values at these

optimal locations in Figure 10. First, we notice that in the multi-camera setup, all depth hypotheses have valid projections. This is important for two reasons. First, this allows all pixel locations and depths to benefit from multi-view depth estimation. Second, it allows the multi-camera setting to exploit larger temporal differences without worrying about non-overlapping regions. To see this, consider the back camera in Figure 10. When considering multiple timesteps (row 1 to row 2), we see that the localization potential dramatically increases for regions where same-camera projections are valid. However, the close-depth regions are unable to make use of the larger temporal differences due to invalid same-camera projection. However, when considering all cameras, we are able to leverage larger temporal differences for both these close-depth and previously invalid regions. In standard stereo, a larger baseline, despite the easier depth estimation, causes large portions of the left & right images to not overlap. However, via our formulation, in multi-timestep temporal stereo with multiple cameras, we can leverage larger temporal differences without worrying about lack of overlap. Finally, we also notice that the multi-camera setup, although better for effective disparity, has much more complex patterns for optimal time difference compared to the single-camera setup. Similar to our conclusions when analyzing rotation, we find that the optimal time difference various for different pixels, cameras, ego-motion, depth, and camera setup.

### B.6 OPTIMAL TIME DIFFERENCE DURING EGO-VEHICLE ROTATION

In this section, we consider the case where theta varies with time. This happens during ego-vehicle turns, and we visualize the optimal time difference over the candidate locations in realistic scenarios of 30, 60, and 90 degree turns in Figure 11. We find that when $\theta$ varies with time, the optimal time difference and optimal projected camera varies wildly over different candidate locations. This shows that it is suboptimal to choose just a few temporal differences for multi-view stereo - a past frame that worked well when the vehicle simply moved forward might fail drastically in more complex ego-motion scenarios such as turns. As such, we conclude it is not only optimal but also necessary to leverage many past timesteps over a long time window for multi-view stereo.

### B.7 ADDITIONAL ANALYSIS ON EFFECTS OF TEMPORAL DIFFERENCE ON MULTI-VIEW DEPTH AMBIGUITY

We find that with the single timestep aggregation used in many methods, less than 20 % of change in object center projection is larger than 1 pixel for objects at 40m - 60m, making accurate multi-view localization impossible. By leveraging 16 past timesteps, we significantly ease multi-view depth estimation (note that for frames with less than 16 timesteps of history, we use as many is available). We do note, however, that the critical front camera is the most difficult view. This is because as seen in Figure 9, points in the front camera can only be projected to itself and are unable to leverage multi-camera depth estimation. However, we find that multi-timestep aggregation can bring % of change in object center projection $> 1$px from 17% and 0.4% to 53% and 22% for objects at 20m-40m and 40m60m, respectively, significantly decreasing the safety risk of front depth estimation. The numerical values of changes in projected location can be seen in Figure 12. As ease of depth estimation isn't simply a binary indicator of "possible" or "not possible", the actual distance between projected locations of two depth hypotheses matters as well. We find that these values increase over various cameras and depths with more temporal aggregation, demonstrating that increased temporal history significantly eases multi-view depth estimation.

## C ADDITIONAL DETAILS FOR HIGH-RESOLUTION, SHORT-TERM DEPTH HYPOTHESIS SAMPLING

In this section, we provide more details for our Gaussian-Spaced Top-k depth candidate sampling method. More specifically, we outline the Gaussian-based down-weighting in greater detail. Let $\sigma$ be a the standard deviation over depth used for Gaussian-base down-weighting, and let $P_l(d)$ as the monocular depth probability of some depth candidate $d$ at sampling iteration $l$. Note that the monocular depth probability is updated over multiple iterations via our down-weighting method.

Suppose $d_l$ is the depth candidate chosen at sampling iteration $l$. Write $P_{\mathcal{N}(\mu,\sigma)}(x)$ as the probability mass at location $x$ in a normal distribution with mean $\mu$ and standard deviation $\sigma$. Then, the

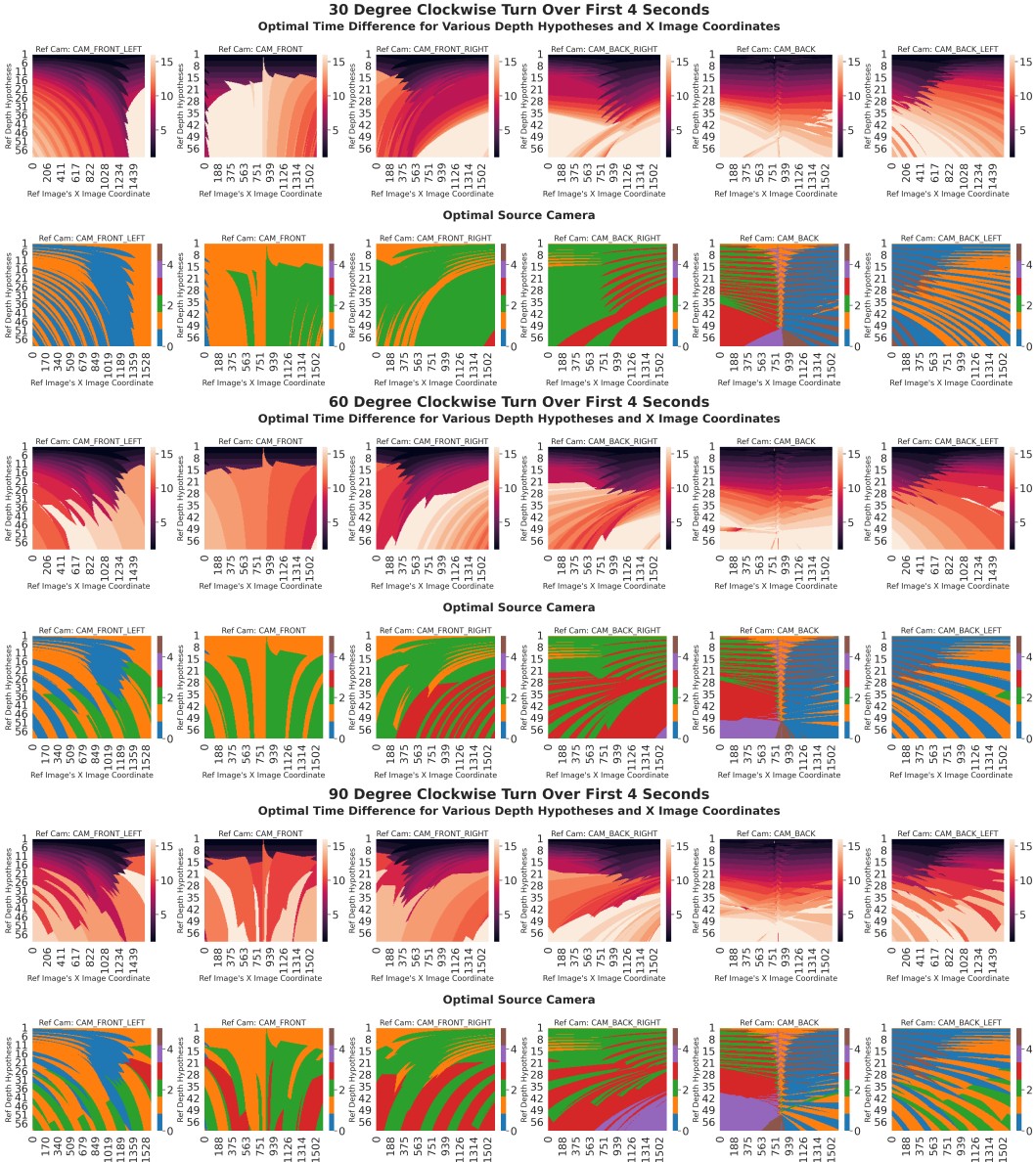

Figure 11: Visualization of the optimal time difference for every candidate location when the ego-vehicle turns 30, 60, or 90 degrees over the first 8 timesteps.

monocular depth probability at some depth location $d$ is updated to be:

$$P_{l+1}(d) = P_l(d) * (1 - P_{\mathcal{N}(d_l,\sigma)}(d) * \sigma\sqrt{2\pi}) \tag{5}$$

This down weights the monocular probability at $d$ by normalized inverse Gaussian factor, with the decrease being larger the closer $d$ is to $d_l$. We find that this simple formulation is enough to force the model to choose depth hypotheses that maintain an effective trade-off between "exploiting" the monocular prior and "exploring" other depth candidates.

# D    ADDITIONAL DETAILS FOR EXPERIMENTAL SETTING

## D.1    MODEL DETAILS

We adopt state-of-the-art BEVDepth (Li et al., 2022c) as our baseline model and conduct experiments with ImageNet-1k pretrained ResNet50 and ResNet101 (He et al., 2016) backbones (He et al., 2016) for validation set comparison. For the test set submission, we use a ConvNeXt-B (Liu et al.,

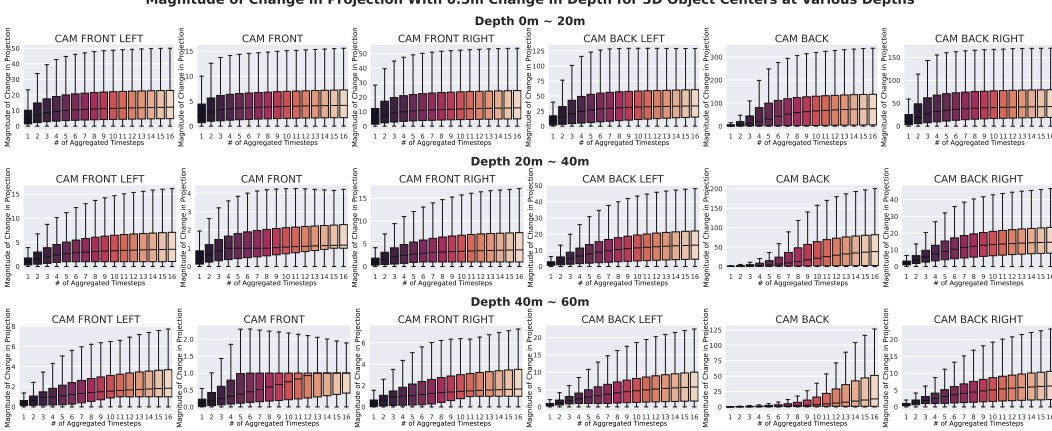

Figure 12: Visualization of absolute value of change in projected location for object centers induced by a 0.5m change in depth.

2022d) CNN backbone pretrained on ImageNet-1k, using pre-trained weights[4] from the mmclassification repository[5], which in-turn are from the official ConvNeXt repository[6]. We use $T = 16$ timesteps for long-term fusion and $k = 7$ depth hypotheses for short-term fusion. For short-term, high-resolution stereo matching, we use a small FPN similar to MaGNet and GWCNet (Bae et al., 2022; Guo et al., 2019) to reduce the matching channel dimension to 64. Group correlation (Guo et al., 2019) is used for matching. For BEV pooling, we adopt the fast implementation from BEV-Fusion (Liu et al., 2022c). We adopt the standard augmentation strategies used by BEVDet (Huang et al., 2021)

### D.2 TRAINING SPECIFICATIONS

All models were trained with the AdamW (Loshchilov & Hutter, 2019) optimizer with weight decay 1e-2 and betas 0.9 and 0.999. We used gradient clipping with max normalization 5 and mixed precision training. Following prior work (Li et al., 2022c; Wang et al., 2022e; Li et al., 2022b), we maintained a constant learning rate for all models and used EMA. Models with a ResNet backbone were trained with 2e-4 learning rate and 64 batch size and our test set model with ConvNeXt was trained with 7.5e-5 learning rate and 24 batch size. For specific details on ConvNeXt training, we simply followed ConvNeXt's setup for object detection. Similar to NeuralRecon (Sun et al., 2021), all long-term fusion models are initially trained for 6 epochs with only one frame, then the weights are loaded into the 16-frame model and trained for 18 more epochs (fairly training for 24 epochs total for the non-CBGS setting).

During both training and inference, we save past BEV feature maps and use them for later timesteps through a sequential dataloader (randomizing over sequences but iterating sequentially through frames within each sequence), keeping our pipeline efficient despite using long-term temporal fusion. As our method processes frames sequentially, the commonly used CBGS (Zhu et al., 2019) training scheme is not readily applicable to our framework. As such, to compare with methods that use CBGS, we simply increase the number of training iterations to match the CBGS cycle without other changes. We emphasize our setting is a *significantly disadvantaged* setting as CBGS is known to substantially boost performance in rarer categories.

## E    NUSCENES TEST SET RESULTS

The `test` set results of SOLOFusion are presented in Table 7. At time of submission, we rank first on the public test set leaderboard without extra data or test-time augmentation.

---

[4]https://download.openmmlab.com/mmclassification/v0/convnext/convnext-base_3rdparty_32xb128-noema_in1k_20220222-dba4f95f.pth

[5]https://github.com/open-mmlab/mmclassification

[6]https://github.com/facebookresearch/ConvNeXt

Table 7: Comparison on the nuScenes `test` set. Extra data is depth pretraining (Guizilini et al., 2020).

| Methods | Backbone | Image Size | Extra Data | Test-Time Aug | mAP↑ | NDS↑ | mATE↓ | mASE↓ | mAOE↓ | mAVE↓ | mAAE↓ |
|---------|----------|------------|------------|---------------|------|------|-------|-------|-------|-------|-------|
| FCOS3D | R101-DCN | 900 × 1600 | ✗ | ✓ | 0.358 | 0.428 | 0.690 | 0.249 | 0.452 | 1.434 | 0.124 |
| DETR3D | V2-99 | 900 × 1600 | ✓ | ✓ | 0.412 | 0.479 | 0.641 | 0.255 | 0.394 | 0.845 | 0.133 |
| UVTR | V2-99 | 900 × 1600 | ✓ | ✗ | 0.472 | 0.551 | 0.577 | 0.253 | 0.391 | 0.508 | 0.123 |
| BEVFormer | V2-99 | 900 × 1600 | ✓ | ✗ | 0.481 | 0.569 | 0.582 | 0.256 | 0.375 | 0.378 | 0.126 |
| BEVDet4D | Swin-B | 900 × 1600 | ✗ | ✓ | 0.451 | 0.569 | 0.511 | **0.241** | 0.386 | 0.301 | **0.121** |
| PolarFormer | V2-99 | 900 × 1600 | ✓ | ✗ | 0.493 | 0.572 | 0.556 | 0.256 | 0.364 | 0.439 | 0.127 |
| PETRv2 | GLOM-like | 640 × 1600 | ✗ | ✗ | 0.512 | 0.592 | 0.547 | 0.242 | 0.360 | 0.367 | 0.126 |
| BEVDepth | ConvNeXt-B | 640 × 1600 | ✗ | ✗ | 0.520 | 0.609 | 0.445 | 0.243 | **0.352** | 0.347 | 0.127 |
| BEVStereo | V2-99 | 640 × 1600 | ✓ | ✗ | 0.525 | 0.610 | **0.431** | 0.246 | 0.358 | 0.357 | 0.138 |
| SOLOFusion | ConvNeXt-B | 640 × 1600 | ✗ | ✗ | **0.540** | **0.619** | 0.453 | 0.257 | 0.376 | **0.276** | 0.148 |

Table 8: Runtime and improvement of methods that utilize short-term temporal stereo with ResNet50. * FPS and Memory are based on our reproduction. Although STS does not provide details, we reduce the matching feature dimension, as SOLOFusion does, for fair comparison. †Only the short-term fusion module is used here. Runtime is measured with a single A6000 GPU.

| Method | FPS | Memory | ΔmAP | ΔmATE |
|--------|-----|--------|------|-------|
| BEVStereo | 1.8 | 4.8 GB | 1.9 | 4.8 |
| STS* | 4.9 | 5.5 GB | 2.4 | 4.4 |
| SOLOFusion† | 12.2 | 3.3 GB | 2.2 | 5.2 |

# F   RUNTIME ANALYSIS OF SOLOFUSION.

In this section, we compare the runtime and memory costs of SOLOFusion with STS and BEVStereo, two prior works that also leverage short-term stereo for 3D detection. The results, as well as each method's improvement over the baseline, are shown in Table 8. Although STS has a slightly larger improvement in mAP, SOLOFusion has runs 2.5x faster and has a larger improvement in localization. Compared to the short-term temporal fusion methods used in existing works, our proposed module is straightforward and efficient with strong performance.

# G   IMPACT OF LONG-TERM TEMPORAL FUSION ON STATIC AND MOVING OBJECTS

In this section, we analyze the effects of long-term temporal fusion separately on static and moving objects. For a thorough analysis, we separate classes into "static" and "movable", referring to classes whose objects move in the nuScenes dataset and classes whose objects do not move. Static classes include construction vehicle[7], traffic cone, and barrier, and movable classes include car, truck, bus, trailer, pedestrian, motorcycle, and bicycle. We then examine the impact of long-term temporal fusion for static objects not in motion, movable objects not in motion, and movable objects in motion. The results are in Figures 13, 14, and 15, respectively. For fair presentation, as *any* amount of temporal fusion already dramatically improves results, the plots start with 1 past frame fused ($T = 1$).

## G.1   OBJECTS NOT IN MOTION

First, we focus on Figures 13 and 14. We observe that for objects not in motion, long-term temporal fusion significantly improves mAP, mATE, and AR@500, which evaluate precision-recall, object center translational error, and average recall, respectively. The significant improvements in mAP and mATE support our theoretical analysis in the main text that long-term fusion can improve localization potential. mAVE, measuring velocity error, does not significantly change as it is already very low for static objects (around 0.05 m/s).

---

[7]We note that we consider construction vehicles as "static" because Figure 8 of the nuScenes (Caesar et al., 2020) paper shows that only around 5% of construction vehicles in the dataset are in motion. As this is already a rare class, there are scarcely any moving construction vehicles in the nuScenes validation set, so we exclude them from the moving object analysis.

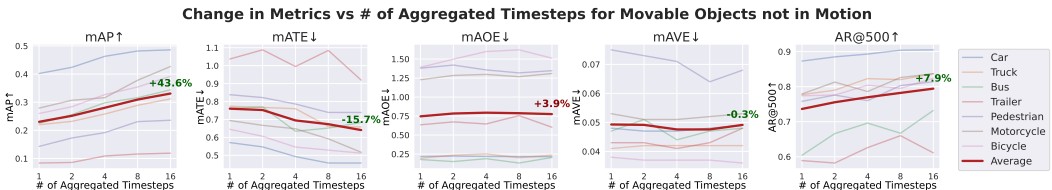

Figure 13: Effects of long-term on static objects not in motion. AR@500 is average recall, averaged over center distance thresholds 0.5, 1.0, 2.0, and 4.0, for 500 predictions per scene, which is standard for nuScenes. Note that traffic cone does not have orientation, so its mAOE is always 0.

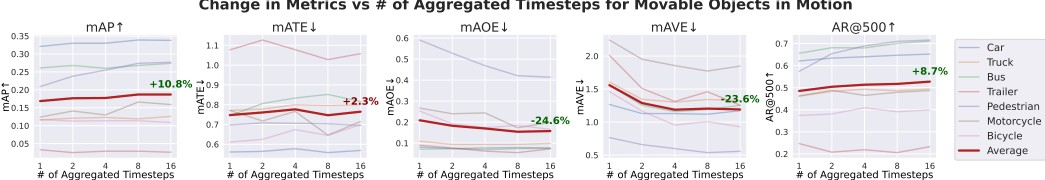

Figure 14: Effects of long-term on movable objects not in motion.

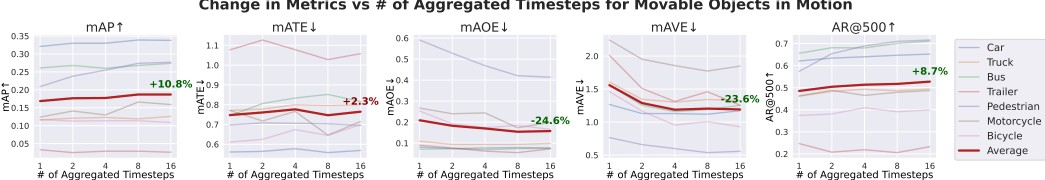

Figure 15: Effects of long-term on movable objects in motion.

We do note, however, that mAOE, measuring orientation error, seems to increase over timesteps. We hypothesize that this increase in mAOE error comes from our training setup. When training multi-timestep models, we use a sequential dataloader as mentioned in Section D, saving BEV features from past timesteps and training iterations to be used for the current timestep and training iteration. However, the image and BEV augmentations for past and current timesteps are different, and although these augmentations are reversed when warping BEV feature maps to the current timestep, this introduces additional noise that might make learning orientation in BEV more difficult. However, we still decide to keep such a sequential dataloader, because saving past BEV features instead of re-predicting them each time during training speeds up the training cycle by around 10x, and simply training the model for longer with faster iterations can more than negate these small drops (see CBGS results in Table 2). Further, the dramatic improvements in all other metrics support the utility of our proposed method.

## G.2 OBJECTS IN MOTION

Next, we examine the impact of long-term fusion on movable objects in motion in Figure 15. We observe significant increases in mAP, mAOE, mAVE, and AR@500 with more timesteps. The dramatic improvements in mAOE and mAVE tell us that our proposed method, by observing moving objects for longer, can better predict their orientation (direction of movement) and their velocity. Notably, AR@500 increases by 8.6%, which is more than the 7.9% increase we observed for movable objects not in motion. We also observe that the mATE for moving objects increases by 2.3% with more timesteps. The slight increase in mATE tells us that although our method significantly improve localization of stationary elements, it does not necessarily benefit precise localization of moving objects.

However, this does not mean that our proposed method does not benefit detection of moving objects. First, we clarify that mATE is a "true-positive" metric (Caesar et al., 2020), meaning it is the average error for *ground-truth matched* predictions. For such true-positive metrics, a prediction is considered "matched/detected" if its center is within 2m of its matched ground truth. We observe that average recall increases with more timesteps, meaning that more predictions are matched with ground truth

Figure 16: Effects of long-term on movable objects excluding pedestrian in motion.

and thus are considered for the true-positive metric. These objects only detected with more timesteps are more difficult and potentially have poorer localization. Thus, although recall is higher, the true-positive metrics, especially mATE which there isn't a clear improvement in, are penalized[8]. Evaluating up to the recall level achieved with fewer timesteps, the increase in mATE is instead +0.8%, which although still not an improvement, is a more accurate picture of the impact of long-term temporal fusion on mATE for moving objects.

Second, we comment that not only does average recall (AR@500) increase, mAP increases as well. This tells us that our proposed method isn't merely scattering a lot of low-confidence boxes, it also meaningfully improves precision-recall. Intuitively, although longer-term fusion isn't able to *precisely* localize moving objects, it can at least better perceive and generally capture their location. Third, the better orientation and velocity estimates for moving object is very important for downstream tasks such as tracking, forecasting, and planning that rely on 3D detection. To verify this, beyond detection metrics, we apply SOLOFusion on the tracking task with a simple greedy-matching algorithm (Yin et al., 2021). Although tracking is not the target of this paper, we achieve 0.540/0.941 AMOTA/AMOTP on the test set, ranking third place (first two entries use sophisticated tracking algorithms). We also believe that this method has huge potential in the field of forecasting and planning as velocity and orientation are the key elements for state estimation. We hope that SOLOFusion can serve as a strong 3D object detection baseline to be used for these downstream tasks.

Finally, looking at change in metrics for individual classes, we observe that pedestrian stands out in Figure 15 as demonstrating huge improvement. This is notable, as pedestrian is a slow-moving class with average velocity 1.3 m/s. This tells us that our long-term fusion module benefits slow-moving objects more than fast-moving ones. We verify that our pipeline does still benefit classes other than pedestrian by plotting change in metrics for other movable objects in motion in Figure 16. The more accurate velocity estimates generated by our pipeline can potentially be used to further improve predictions for moving objects, and we hope to explore such directions in future work.

---

[8]We verified this with a toy example by generating two modified sets of predictions. For the first, we removed predictions for some scenes. For the second, we kept the predictions but randomized velocity predictions for those same scenes, representing a scenario where more objects are captured, but with worse attribute predictions. The latter had significantly worse mAVE than the former using official evaluation code, as expected.

