# OpenReview forum: "Time Will Tell: New Outlooks and A Baseline for Temporal Multi-View 3D Object Detection"
_ICLR.cc/2023/Conference — ICLR 2023 notable top 5%_

### Official Review · Reviewer_FyEa · 2022-10-20

**Confidence:** 5
**Correctness:** 3
**Technical Novelty And Significance:** 3
**Empirical Novelty And Significance:** 4
**Recommendation:** 8

**Clarity, Quality, Novelty And Reproducibility:**

Clarity: Overall good, yet insufficient for network details.

Quality: Overall good.

Novelty: The analysis is novel and solid, but the technical contribution seems incremental.

Reproducibility: The implementation of the temporal setting is easy to reproduce, but the framework is hard to reproduce without further details.

**Details Of Ethics Concerns:**

None.

**Strength And Weaknesses:**

Strengths:

- The basic idea is easy to follow and the main motivation is clear.
- The theoretical and empirical analyses are insightful. They can be important supplements for this line of works in the community.
- The conclusion is clear and the paper proposes an efficient way to leverage both conclusions.
- The proposed method achieves strong experiment results, both compared to its baseline and other SoTA methods.
- Some empirical studies are interesting and important for practical use, such as the analysis about balancing temporal fusion and resolution in Table 7.

Weaknesses:

- The analysis part is a little wordy (although I understand it is fruitful, there is no need to clarify such simple conclusions with so many paragraphs) while the methodology part is so brief, even without a figure showing the details of networks. The re-organization of these contents in the main paper and supplemental materials should be considered.
- The technical contribution seems incremental (though it may be related to the presentation problems). The framework seems a simple combination of BEVDepth and BEVStereo/DfM while only adjusting some settings for temporal aggregation. The author should clarify this point more clearly, and add at least a specific figure showing the network architecture as well as the distinguished technical design details.
- As the involved timestep increases, another problem is that the effect of object motion on the stereo matching is more notable. Analysis for this point is missing in this paper. Such problems can severely and intrinsically challenge the conclusion that "long-term matching" is better.
- (Minor) It would be better to have more references at:
1. In the first paragraph and the conclusion section, the author mentions that "depth estimation is the main bottleneck of camera-only works". Although it can be common sense for researchers from this community, it would be better to include some references to support such claims.
2. In Table 1, it would be better to compare SOLOFusion with a missing reference about the discussion of long-range, "UniFormer: Unified Multi-view Fusion Transformer for Spatial-Temporal Representation in Bird's-Eye-View, Arxiv 2022". Besides, DfM is missing in the MVS part and there is another line of work that is perpendicular to LSS-based, like OFTNet, ImVoxelNet, and MV-FCOS3D++ (which do not predict depth probability for 2D-3D lifting, among them only the last one has temporal modeling).
- (Minor) Minor typos/grammatical mistakes:
1. Footnote1, "because it while it is"
2. The caption of Figure 4 seems to be covered by Figure 5.

**Summary Of The Paper:**

This paper makes a systematic study in terms of the critical factors in temporal stereo matching for camera-only 3D object detection and proposes corresponding solutions with a framework, SOLOFusion, to address the problems. It concludes that the limited history usage and low granularity of matching resolution are the two bottlenecks for previous methods, among which the former is more important and can compensate for the latter to some extent. The resulting solution leverages both low-resolution, long-term and high-resolution, short-term information to construct temporal multi-view features effectively and efficiently. It achieves a new SoTA on the nuScenes benchmark. The ablation studies also support the analysis.

**Summary Of The Review:**

This paper formulates the problem of camera-only 3D detection from videos from a temporal multi-view stereo perspective and provides an analysis based on the "localization potential" concept. It results in critical conclusions that the usage of long-term histories and high-resolution features for matching are the most important factors in this setting and proposes corresponding solutions. Experiments show the effectiveness of the proposed approaches and demonstrate their importance in real-time practical use. The main problem is focused on the organization of contents in the main paper and supplemental materials, the incremental technical contribution for framework design, and the missing discussion about object motion. I would recommend weak acceptance at the current stage because it is an important supplement for this line of work and believe it can bring new insight to this community.

---

> ### Author Response · Authors · 2022-11-15
> **Response to Reviewer FyEa [1/2]**
>
> We thank the reviewer for their feedback, and we appreciate that they found the empirical analysis insightful and important for the community.
>
> >”The analysis part is a little wordy (although I understand it is fruitful, there is no need to clarify such simple conclusions with so many paragraphs) while the methodology part is so brief, even without a figure showing the details of networks. The re-organization of these contents in the main paper and supplemental materials should be considered.”
>
> >“The technical contribution seems incremental (though it may be related to the presentation problems). The framework seems a simple combination of BEVDepth and BEVStereo/DfM while only adjusting some settings for temporal aggregation. The author should clarify this point more clearly, and add at least a specific figure showing the network architecture as well as the distinguished technical design details.”
>
> We have re-written parts of the analyses and to make it more clear. Re-shuffling some content between the main text and appendix, we have added a figure to the main text showing the model architecture and details which we hope clarifies our method and contributions. We briefly summarize them below.
>
> First, in our work, we systematically and theoretically analyze that the long-term fusion can increase localization potential and can offset low-resolution matching, resulting in a tradeoff between timesteps and feature granularity. We develop a new paradigm of long-term low-resolution and short-term high-resolution, which is a natural extension of our theoretical analysis instead of a tuning of parameters on existing works. We believe that our careful study of the ability of long-term fusion to offset low-resolution matching is new and important for the community.
>
> In addition, we propose a new Gaussian-Spaced Top-k sampling strategy for our short term high resolution module. This strategy substantially improves performance while remaining efficient as shown in Table 4 and further benefits our already strong long-term model in Table 5. Finally, compared to other approaches for short-term fusion as in BEVStereo and STS, the proposed sampling strategy is more efficient as analyzed in Table 8.
>
> Our final model is intuitive, but it demonstrates good empirical results and supports our theoretical conclusions through extensive ablations. We hope that SOLOFusion can be a clear, but strong, baseline for the community to explore temporal fusion for camera-only 3D detection.
>
> >“As the involved timestep increases, another problem is that the effect of object motion on the stereo matching is more notable. Analysis for this point is missing in this paper. Such problems can severely and intrinsically challenge the conclusion that "long-term matching" is better.”
>
> We have added extensive analyses for the effects of long-term matching on both static and moving objects in Appendix G. We summarized our findings in “Overall Response to Reviewers” above, but we recommend reviewers reference the plots in the appendix.
>
> >”(Minor) It would be better to have more references at:”
>
> >”In the first paragraph and the conclusion section, the author mentions that "depth estimation is the main bottleneck of camera-only works". Although it can be common sense for researchers from this community, it would be better to include some references to support such claims.”
>
> We have added references to two works that perform oracle experiments, replacing predicted depth with GT depth, to support these claims: “These long-term past observations are critical for better depth estimation, which has been demonstrated through oracle experiments (Wang et al., 2021b; Jing et al., 2022) as the main bottleneck of camera-only pipelines due to their lack of explicit depth measurements.”
>
> >”In Table 1, it would be better to compare SOLOFusion with a missing reference about the discussion of long-range, "UniFormer: Unified Multi-view Fusion Transformer for Spatial-Temporal Representation in Bird's-Eye-View, Arxiv 2022". Besides, DfM is missing in the MVS part and there is another line of work that is perpendicular to LSS-based, like OFTNet, ImVoxelNet, and MV-FCOS3D++ (which do not predict depth probability for 2D-3D lifting, among them only the last one has temporal modeling).”
>
> We have added these references to the related work section and Table 1. Notably, we mention that although UniFormer does perform experiments with up to 10 timesteps (5 seconds) of temporal aggregation, their pipeline peaks in performance at 6 timesteps (3 seconds) and they adopt this time window for their final model. Our proposed method instead evaluates on 3D object detection (which requires detection of moving elements unlike map segmentation in UniFormer), and demonstrates improvement up to 16 timesteps (8 seconds).

---

> > ### Author Response · Authors · 2022-11-15
> > **Response to Reviewer FyEa [2/2]**
> >
> > >”(Minor) Minor typos/grammatical mistakes:
> > Footnote1, "because it while it is"
> > The caption of Figure 4 seems to be covered by Figure 5.”
> >
> > We thank the reviewer for catching these errors, and we have fixed these mistakes in the updated manuscript.
> >
> > >”Reproducibility: The implementation of the temporal setting is easy to reproduce, but the framework is hard to reproduce without further details.”
> >
> > We plan to release code soon, and to facilitate reproduction and understanding of the paper, Appendix D has been expanded to include more implementation & training details. We hope that this allows the community to better understand and use our work.

---

> > > ### Comment · Reviewer_FyEa · 2022-12-04
> > > **Post-Rebuttal Comments**
> > >
> > > Thanks for the response from the authors. Most of my concerns are addressed to some extent and I would raise the rating to 8. I believe this paper is an insightful work and will provide solid discussions for reference for this community.

---

### Official Review · Reviewer_U6zw · 2022-10-24

**Confidence:** 4
**Correctness:** 4
**Technical Novelty And Significance:** 3
**Empirical Novelty And Significance:** 3
**Recommendation:** 8

**Clarity, Quality, Novelty And Reproducibility:**

Clarity:
Accaptable but could be improved. See "weaknesses"

Quality:
Good. I will rank this work as the top 20% in the area of 3D object detection.

Novelty:
Good. This is one of the few early attempts that successfully employ multi-view stereo for 3D object detection.

Reproducibility:
Good. The proposed method is simple and easy to reproduce.

**Strength And Weaknesses:**

Strength:
+ This work borrows a lot of knowledge and tools from multi-view stereo to help us understand and analysis how temporal context should be used for vision-based 3D object detection. Introducing multi-view stereo to 3D object detection is a valuable direction that probably worth much more research, and to my knowledge this work can be seen as one of the few works that open up this direction.
+ The analysis based on the proposed "localization potential" well-explained the motivation of using both long-term and short-term memory, and the conclusions shown in Fig.4-6 are quite interesting and inspiring.
+ The final method is quite simple, easy to be re-produced.
+ Experimental results are very promising,  refreshing previous state-of-the-art by a significant margin, showing the importance of using long-temporal context.
+ Thorough abation study.

Weaknesses:
- I think the biggest issue is the presentation. I appreciate the dense technical content, which indeed brings difficulty in clarification; But the presentation could definitly be improved. For instance, a reader who has weak background on multi-view stereo may be confusing when discussing the relationship between multi-vew stereo and 3D object detection, since there is little preliminary knowledge introduced in the main text.  Maybe it would be better to elaborate more on background, and defer other stuff to the appendices.



**Summary Of The Paper:**

This paper investigate how temporal context should be used for vision-based 3D object detection. They treat temporal 3D object detection as a multi-view stereo problem, where history frames can be seen as multiple views. A criterion termed as "localization potential" is defined to reflect the level of difficluty for multi-view stereo, upon which they did in-depth analysis and concluded that using long-term, low-resolution temporal context benefits detection.

Based on the analysis, a new 3D object detection method named SOLOFusion is proposed. The core idea is using both long-term and short-term memory. The long-term memory stores low-res features, and is fused by BEV cost volume; the short-term memory stores high-res features from the very last frame, and is fused  by plane-sweep cost volume. Experimental results show the proposed method can serve as a new strong baseline. Despite its simplicity, the results are quite impressive, refreshing previous state-of-the-art by a significant margin.

**Summary Of The Review:**

In general, this is a good paper that worths acceptance.
My biggest concern is the presentation so I give my recommendation as "6: marginally above the acceptance threshold". If the presentation could be improved, I will be willing to raise my rating to "8: accept, good paper".

---

> ### Author Response · Authors · 2022-11-15
> **Response to Reviewer U6zw**
>
> We thank the reviewer for their review, and we are glad that they found the connection between multi-view stereo and camera-only 3D detection interesting.
>
> >”I think the biggest issue is the presentation. I appreciate the dense technical content, which indeed brings difficulty in clarification; But the presentation could definitely be improved. For instance, a reader who has weak background on multi-view stereo may be confusing when discussing the relationship between multi-view stereo and 3D object detection, since there is little preliminary knowledge introduced in the main text. Maybe it would be better to elaborate more on background, and defer other stuff to the appendices.”
>
> We have reshuffled parts of the main text and appendix to make space for more background information in the main text. In addition to re-writing portions of the text to make key takeaways more clear, we added Section 3.1 “Background of Multi-View Stereo Matching”:
>
> “In this section, we give additional background on the task of multi-view stereo. At a high level, multi-view stereo matching estimates the depth mAP of an image (reference view) by leveraging additional images taken of the same scene (source views). For each pixel in the reference view, multi-view stereo frameworks consider for that pixel many possible depth hypotheses (Yao et al., 2018) - locations that this pixel could be in the 3D world. Of these hypotheses, the true depth at which that pixel exists would appear the most similar through the reference and source views. Thus, each depth hypothesis is projected onto the source views, and the depth with the best image feature match between the source view projection and the reference pixel is selected as the depth estimate. In the standard stereo setting, the reference and source views are left and right cameras with synchronized, rectified images. For temporal stereo, the reference and source views are images captured at the current and past timesteps as illustrated in Figure 1. For additional details on multi-view stereo, we refer readers to MVSNet (Yao et al., 2018).”
>
> Additional details of these changes are in the “Overall Response to Reviewers” above.

---

### Official Review · Reviewer_Bu5e · 2022-10-25

**Confidence:** 3
**Correctness:** 4
**Technical Novelty And Significance:** 4
**Empirical Novelty And Significance:** 4
**Recommendation:** 8

**Clarity, Quality, Novelty And Reproducibility:**


Clarity, Quality, Novelty:

As mentioned in the strengths, this paper is organised well and has several novel contributions. Overall, this is a high-quality paper.

Reproducibility:

From Section 5, it seems not very difficult to reproduce the paper. However, we are still look forward to the officially released code.

**Strength And Weaknesses:**

Strengths:

1. New formulation of the exiting multi-view, multi-frame image based 3D detection methods.

2. Use both theoretical and empirical analysis by introducing localisation potential, to unveil the importance of longer time horizon.

3. Impressive performance in nuScenes leaderboard.

4. Writing is clear and easy to follow.

Weakness:

I have one question regarding the experimental analysis. Did the authors analyse the performance improvement for objects with different speed? It is interesting to see the impact of long term fusion on static, slowly moving and fast moving objects.

**Summary Of The Paper:**

In this paper, the authors study vision only 3D object detection for autonomous driving. First, the authors formulate the recent multi-frame multi-view methods as temporal stereo matching. Then localisation potential is proposed and analysed theoretically and empirically to show the necessity of larger time window. To handle the efficiency of long term computation, coarser resolution feature maps are used in the long term and finer resolution in the short term. In the experiments, the proposed method outperforms others in nuScenes leaderboard.

**Summary Of The Review:**


I would like to accept this paper. If the authors can provide more analysis for the method (for example, the question I have in the weakness), I tend to further increase my rating.

---

> ### Author Response · Authors · 2022-11-15
> **Response to Reviewer Bu5e**
>
> We thank the reviewer for their review, and we appreciate that they found our proposed framework and analysis novel and useful.
>
> >”I have one question regarding the experimental analysis. Did the authors analyze the performance improvement for objects with different speeds? It is interesting to see the impact of long term fusion on static, slowly moving and fast moving objects.”
>
> We have added analyses on the impact of long-term fusion on static and moving objects in Appendix G. We summarized our findings in “Overall Response to Reviewers” above, but we recommend reviewers reference the plots in the appendix.
>
> >”From Section 5, it seems not very difficult to reproduce the paper. However, we are still looking forward to the officially released code.”
>
> We plan to release code soon, and we have added additional implementation details to Appendix D.

---

### Author Response · Authors · 2022-11-15
**Overall Response to Reviewers [1/2]**

We sincerely thank the reviewers for their thoughtful comments and feedback. We appreciate that all reviewers agreed that the theoretical and empirical analyses of localization potential were insightful and that the conclusions and the strong baseline are valuable for the community. We address the main reviewer concerns below, and we have uploaded an updated manuscript. Please note that just for this rebuttal version, the significant manuscript additions are in blue for reviewer convenience.

**1. The presentation of the paper can be improved, and authors should consider reshuffling materials between the main text and supplementary.**

Reviewer U6zw cited concerns that although the technical content is useful, there is not enough preliminary information on multi-view stereo prior to connecting multi-view stereo with 3D detection. Reviewer RyEa mentioned that the analysis was a bit wordy and suggested model details should be further clarified in the main text.

We have improved the manuscript for clarity. Specifically, we added Section 3.1 “Background of Multi-View Stereo Matching” prior to describing the unified stereo + camera 3D detection formulation to better motivate the connection for readers potentially less familiar with multi-view stereo. Further, we have rephrased and pruned parts of the analysis to make it and the main takeaways more clear. Finally, we have included the model architecture diagram (Figure 7) in the main text to make our method more clear. We note that we have moved the test set results to the appendix.

**2. The effects of long-term matching on moving objects should be analyzed.**

Reviewer Bu5e raised a question regarding the effects of long-term fusion on moving objects, and reviewer FyEa cited concerns that with larger motion over long-term fusion, object movement can potentially inhibit stereo matching and asked for analysis of this point.

We agree that evaluating the effects of long-term fusion on static vs moving objects is important, and we have updated the supplementary with detailed analysis regarding this point in Appendix G. We encourage the reviewers to reference the plots and analysis in this section, but we summarize our findings here. For fair presentation, as _any_ temporal fusion already improves results, our plots and analyses start with 1 past frame fused ($T=1$).

We first split the classes in nuScenes into “static” (objects that don’t move: construction vehicle, traffic cone, barrier) and “movable” (objects that can move: car, truck, bus, trailer, pedestrian, motorcycle, bicycle) and separately evaluated long-term fusion on “static objects not in motion,” “movable objects not in motion,” and “movable objects in motion”.

Considering objects **not in motion** (static and movable), we observe that long-term fusion significantly improves mAP$\uparrow$ (precision-recall), mATE$\downarrow$ (translation error), and AR@500$\uparrow$ (average recall). mAVE$\downarrow$ (velocity error) for objects not in motion is already low, so there is little change. We find that mAOE$\downarrow$ (orientation error) does seem to increase: +13.3% for static objects, mostly fueled by the rare construction vehicle class, and 3.9% for movable objects. We attribute this increase to our training setup, which uses different image & BEV augmentations (such as rotation and flip) for every training iteration, causing the past timestep BEV features, taken from past training iterations, to have different augmentations (albeit inverted before fusion) as those of the current timestep. However, such a sequential dataloader speeds up training time by 10x, and simply training for longer on faster iterations more than compensates for this slight drop. Overall, the dramatic improvements in all other metrics support the utility of our proposed method.

Next, we consider movable objects **in motion**. We observe significant improvements in mAP$\uparrow$, mAOE$\downarrow$, mAVE$\downarrow$, and AR@500$\uparrow$ with more timesteps. By observing objects for longer, the proposed method can better predict orientation and velocity. Notably, AR@500$\uparrow$ is higher for movable objects in motion than not in motion (+8.7% vs +7.9%), and the +10.8% improvement in mAP$\uparrow$ (precision-recall metric) tells us improvements in AR@500$\uparrow$ are not from a plethora of low-confidence boxes scattered everywhere.

---

> ### Author Response · Authors · 2022-11-15
> **Overall Response to Reviewers [2/2]**
>
> We do observe that mATE$\downarrow$ does increase slightly (+2.3%) with more timesteps. However, we clarify that mATE$\downarrow$ is a “true-positive” metric, computed over predictions matched with GT. As using more timesteps matches more predictions with GT (+8.7% AR@500$\uparrow$), mATE$\downarrow$ is computed with these additionally matched predictions, which are of harder objects missed with fewer timesteps. When we instead evaluate mATE$\downarrow$ up to recall achieved with fewer timesteps, the increase in mATE$\downarrow$ is +0.8%, which although still not an improvement, is a more accurate picture of the impact of long-term temporal fusion on mATE$\downarrow$ for moving objects.
>
> However, we comment that long-term fusion does improve average recall (AR@500$\uparrow$) and mAP$\uparrow$. From this, intuitively, although longer-term fusion isn't able to _precisely_ localize moving objects, it can at least better perceive and generally capture their location. We believe that the improvement in other metrics (mAP$\uparrow$, mAOE$\downarrow$, mAVE$\downarrow$, AR@500$\uparrow$) for moving objects still demonstrates the utility of long-term fusion, not to mention the improvement for non-moving objects. To verify this, beyond detection metrics, we apply SOLOFusion on the tracking task with a simple greedy-matching algorithm. Although tracking is not the target of this paper, we achieve 0.540/0.941 AMOTA/AMOTP on the test set, ranking third place (first two entries use sophisticated tracking algorithms). We also believe that this method has huge potential in the field of forecasting and planning as velocity and orientation are the key elements for state estimation.
>
> Finally, we observe that long-term fusion especially helps moving pedestrians, which are slow-moving and also critical entities, showing that long-term fusion does help more for slower-moving objects. We believe the more accurate velocity estimation from long-term fusion can be looped back to further improve perception of fast moving objects (as a more accurate prior for where objects are in the next time step), and we hope to explore this in future work.
>
> **3. More model and implementation details are needed for reproducibility.**
>
> We plan to release code for this work soon, and we have added a lot more implementation details to Section D of the supplementary so readers can better understand the work. We have detailed both our validation set and test set settings.

---

### Decision · Program_Chairs · 2023-01-20

**Decision:**

Accept: notable-top-5%

**Justification For Why Not Higher Score:**

NA

**Justification For Why Not Lower Score:**

This paper is a clear accept with 3x accept, good paper.

**Metareview: Summary, Strengths And Weaknesses:**

This paper is a clear accept with 3x accept, good paper. The reviews are mostly positive. Particularly, a new formulation of the exiting multi-view, multi-frame image based 3D detection methods. It shows excellent performance on the nuScenes leaderboard. The writing is clear and easy to follow. The theoretical and empirical analyses are insightful. They can be important supplements for this line of works in the community. The weaknesses are mostly minor that can be easily corrected in the final version.

**Note From Pc:**

if the above contains the word "oral" or "spotlight" please see: "oral" presentation means -> notable-top-5% and "spotlight" means -> notable-top-25%. As stated in our emails, we are disassociating presentation type from AC recommendations